# Structural insights on ligand recognition at the human leukotriene B4 receptor 1

Nairie Michaelian[1,2], Anastasiia Sadybekov[1,2], Élie Besserer-Offroy[3,4], Gye Won Han[1,2], Harini Krishnamurthy[5], Beata A. Zamlynny[5], Xavier Fradera[5], Phieng Siliphaivanh[5], Jeremy Presland[5], Kerrie B. Spencer[5], Stephen M. Soisson[5], Petr Popov[6,7], Philippe Sarret[3], Vsevolod Katritch[1,2,8] & Vadim Cherezov[1,2,7 ✉]

The leukotriene B4 receptor 1 (BLT1) regulates the recruitment and chemotaxis of different cell types and plays a role in the pathophysiology of infectious, allergic, metabolic, and tumorigenic human diseases. Here we present a crystal structure of human BLT1 (hBLT1) in complex with a selective antagonist MK-D-046, developed for the treatment of type 2 diabetes and other inflammatory conditions. Comprehensive analysis of the structure and structure-activity relationship data, reinforced by site-directed mutagenesis and docking studies, reveals molecular determinants of ligand binding and selectivity toward different BLT receptor subtypes and across species. The structure helps to identify a putative membrane-buried ligand access channel as well as potential receptor binding modes of endogenous agonists. These structural insights of hBLT1 enrich our understanding of its ligand recognition and open up future avenues in structure-based drug design.

[1] Bridge Institute, USC Michelson Center for Convergent Bioscience, University of Southern California, Los Angeles, CA, USA. [2] Department of Chemistry, University of Southern California, Los Angeles, CA, USA. [3] Department of Pharmacology-Physiology, Faculty of Medicine and Health Sciences, Institut de Pharmacologie de Sherbrooke, Université de Sherbrooke, Sherbrooke, QC, Canada. [4] Department of Molecular and Medical Pharmacology, David Geffen School of Medicine, University of California at Los Angeles, Los Angeles, CA, USA. [5] Merck Research Laboratories, Merck & Co., Inc., Kenilworth, NJ, USA. [6] Center for Computational and Data Intensive Science and Engineering, Skolkovo Institute of Science and Technology, Moscow, Russia. [7] Research Center for Molecular Mechanisms of Aging and Age-Related Diseases, Moscow Institute of Physics and Technology, Dolgoprudny, Russia. [8] Department of Quantitative and Computational Biology, University of Southern California, Los Angeles, CA, USA. ✉email: cherezov@usc.edu

Leukotriene B4 (LTB4) is a pro-inflammatory lipid mediator and potent chemoattractant acting via two G protein-coupled receptors (GPCRs), the LTB4 receptors 1 (BLT1) and 2 (BLT2)[1]. hBLT1 was first identified and cloned in 1997 by Yokomizo et al.[2] as a high-affinity LTB4 receptor. hBLT2, which shares a 32% sequence identity with hBLT1, was discovered three years later and is characterized as a low-affinity receptor (20-fold lower affinity for LTB4)[3]. BLT1 is predominantly expressed in leukocytes, while BLT2 has a wider expression profile across multiple cells and tissues. Both receptors couple mainly to the $G_{i/o}$[3–5] as well as to the $G_q$ subfamily ($G_q$, $G_{11}$, $G_{14}$, and $G_{16}$), with BLT1 signaling primarily through $G_{16}$[6] and BLT2 through $G_{14}$[7]. The two receptor subtypes also differ in their ligand recognition, with BLT2 having a broader ligand specificity to endogenous eicosanoids than BLT1[8].

BLT1 regulates inflammation-related processes such as recruitment of neutrophils, monocytes[9,10], and T cells[11,12], as well as smooth muscle cell chemotaxis and proliferation[13,14]. Various studies, including those using BLT1 knockout mice, have associated BLT1 with diseases such as asthma[15], influenza[16], arthritis[9], atherosclerosis[14], diabetes[10], and cancer[17]. Extensive attempts have been made to develop BLT1 ligands for different indications; however, none of them have made it to market. Clinical trials of these agents have been unsuccessful due to adverse side effects, low efficacy, and long plasma half-life[18,19]. Although the underlying cause for this failure has not been established, one explanation may be the off-target activity of BLT1 ligands. Apart from their interactions with BLT2, BLT1 ligands are also known to bind to non-GPCR proteins, such as the peroxisome proliferator-activated receptors PPARα[20] and PPARγ, 5-lipoxygenase (5-LO)[21], and the transient receptor potential channel TRPV1[22]. The knowledge of the hBLT1 structure and the molecular determinants of its ligand recognition may provide additional clues for designing the next generation of selective ligands with improved pharmacological properties.

Recently, a structure of guinea pig BLT1 (gpBLT1) in complex with the inverse agonist BIIL-260 (PDB ID 5X33) has been solved[23] at a resolution of 3.7 Å. To provide a structural basis for hBLT1 ligand recognition and a deeper understanding of its mechanism of action, here we determined a 2.9 Å resolution crystal structure of hBLT1 in complex with a selective antagonist MK-D-046, which was developed for the treatment of type 2 diabetes (T2D) and other inflammatory conditions[24]. The structure determination work was complemented by site-directed mutagenesis and docking studies.

## Results

**Structure determination of hBLT1.** The hBLT1 construct used for structure determination was engineered to facilitate crystallization in lipidic cubic phase (LCP) by truncating N-terminal residues 1–4 and C-terminal residues 311–352, fusing flavodoxin with an arginine-arginine linker in the third intracellular loop (ICL3) between residues 212 and 213, and introducing five thermostabilizing point mutations (L106$^{3.41}$W, S116$^{3.51}$Y, A196$^{5.53}$I, C287$^{7.55}$F, and S310A; superscripts represent Ballesteros-Weinstein nomenclature[25]). In IP$_1$ (myo-inositol 1 phosphate) signaling assays, all five mutations combined decreased the efficacy and potency of the agonist LTB4 (Table 1 and Supplementary Fig. 1a). This reduction was mainly due to the S116$^{3.51}$Y mutation, which restores the conserved DRY motif commonly found in transmembrane helix 3 (TM3) of class A GPCRs[26,27] and likely affects the receptor's interaction with G proteins. All other stabilizing mutations had little or no effect on LTB4 potency or efficacy. In contrast to LTB4, all 5 mutations combined had almost no effect on the potency and inhibition

efficacy of the antagonist MK-D-046 (Table 1 and Supplementary Fig. 2a). The actual hBLT1 crystallization construct (hBLT1-CC) did not respond to LTB4 stimulation likely due to the fused flavodoxin in the ICL3 (ICL3-flav), which was the only modification that completely abolished LTB4 signaling (Table 1 and Supplementary Fig. 1a). Radioligand binding assays confirmed that binding affinities for both LTB4 and MK-D-046 decreased only slightly (<3-fold) at hBLT1-CC compared to wild-type hBLT1 (hBLT1-WT) (Supplementary Table 1 and Supplementary Fig. 3a, b).

The hBLT1 structure in complex with MK-D-046 was solved in an orthorhombic P 2$_1$ 2 2$_1$ space group with one monomer per asymmetric unit at an anisotropic resolution truncated at 2.9, 2.9, 3.6 Å in the a*, b*, c* directions, respectively (Supplementary Table 2 and Supplementary Fig. 4). The crystal packing is typical of any LCP grown crystals with the hBLT1 receptor packed in layers and with the flavodoxin fusion protein mediating the majority of the polar contacts (Supplementary Fig. 5a). Within each layer, receptors engage in both parallel and antiparallel interactions. The crystallographic parallel dimer interface involves TM1 and helix 8 (H8) (Supplementary Fig. 5b).

**Overall structure of hBLT1.** The hBLT1 structure, captured in the inactive state, adopts the canonical seven transmembrane helical (7TM) fold of class A GPCRs (Fig. 1a, b). MK-D-046, a BLT1-selective antagonist (Fig. 1c; cAMP IC$_{50}$: hBLT1 = 2 nM[24], hBLT2 = no ligand response), was modeled in the prominent electron density inside the orthosteric ligand-binding pocket of hBLT1 (Fig. 1d and Supplementary Fig. 4e–g).

BLT1 belongs to the γ branch of class A GPCRs along with the chemokine and opioid receptors, among others[26]. The inactive state of the hBLT1 structure is apparent from its comparison with other γ branch receptor structures, such as the κ-opioid receptor (KOR), which was previously solved in both the inactive (PDB ID 4DJH) and active (PDB ID 6B73) states (Fig. 2a). In particular, conformations of two conserved microswitches[27,28] in hBLT1, NP$^{7.50}$xxY, and P$^{5.50}$-I$^{3.40}$-F$^{6.44}$ (P-V-F in hBLT1), closely resemble those of the KOR inactive state (Fig. 2b, c).

The crystal structures of hBLT1 and gpBLT1 (PDB ID 5X33) are similar in both sequence (73.7% sequence identity) and overall backbone conformation (Cα RMSD > 0.74 Å, over 90% of best matching 7TM residues). There are, however, substantial differences between the structures of the two orthologs in several regions. In gpBLT1, TM7 ends at A287$^{7.56}$ and the amphiphilic H8 is not resolved (Supplementary Fig. 6). In hBLT1, both the intracellular end of TM7, which includes the triple glycine motif (G289$^{7.57}$-G290$^{7.58}$-G291$^{7.59}$), and H8 were modeled. The conformation of H8 in hBLT1, however, may be influenced by crystal packing as well as an artificial sequence (EFLEVLFQ), which follows H8 and consists of the EcoRI site and a portion of the PreScission Protease (PSP) recognition site. This artificial sequence forms an α-helix that folds underneath the receptor's intracellular side (Supplementary Fig. 6) and engages in several polar interactions with native hBLT1 residues.

The extracellular regions also differ between hBLT1 and gpBLT1. While the orthosteric binding pocket of hBLT1 is widely exposed to the solvent at the extracellular side (Fig. 1b), that of gpBLT1 is partially blocked. This blockade is due to R263$^{7.32}$ in gpBLT1, whose hBLT1 equivalent is a smaller residue, S264$^{7.32}$ (Fig. 3b, c, f), that is additionally displaced outward by 2.2 Å due to a kink in the extracellular tip of TM7, compared to the straight TM7 in gpBLT1. The kink in hBLT1 is stabilized by N268$^{7.36}$, which is replaced with K267$^{7.36}$ in gpBLT1. The S$^{7.32}$ and N$^{7.36}$ combination is almost unique to hBLT1, as this pair of residues is conserved in only two other orthologs, the chimpanzee and the

**Table 1 Cell-surface expression and signaling data for hBLT1 mutants evaluated in IP$_1$ production assays.**

| Category | Mutation | Cell-surface expression, % of WT ± SEM ($n$) | EC$_{50}$ LTB4, nM ± SEM ($n$) | $E_{max}$, % ± SEM ($n$) | IC$_{50}$ MK-D-046, nM ± SEM ($n$) | $I_{max}$, % ± SEM ($n$) |
|---|---|---|---|---|---|---|
| Wild type | hBLT1-WT | 100 ± 8 (5) | 0.64 ± 0.11 (7) | 100.0 ± 0.1 (7) | 9 ± 2 (5) | 92 ± 5 (5) |
| Mutants of hBLT1-CC | hBLT1-CC | 114 ± 8 (3) | N/D | N/D | N/A | N/A |
| | 5 mut | 92 ± 7 (3) | 6.1 ± 0.2 (3) | 70 ± 7 (3) | 5 ± 2 (3) | 71 ± 8 (3) |
| | ICL3-flav | 81 ± 9 (3) | N/D | N/D | N/A | N/A |
| | Δ 311–352 | 111 ± 10 (3) | 1.6 ± 0.3 (3) | 102 ± 13 (3) | 20 ± 3 (3) | 94 ± 4 (3) |
| | L106$^{3.41}$W | 125 ± 13 (3) | 1.16 ± 0.12 (3) | 101 ± 10 (3) | 18 ± 3 (3) | 86 ± 6 (3) |
| | S116$^{3.51}$Y | 106 ± 9 (3) | 4.3 ± 0.3 (3) | 75 ± 3 (3) | 10.8 ± 1.9 (3) | 92 ± 9 (3) |
| | A196$^{5.53}$I | 127 ± 13 (3) | 0.35 ± 0.14 (3) | 99 ± 5 (3) | 30 ± 5 (3) | 82 ± 5 (3) |
| | C287$^{7.55}$F | 89 ± 12 (3) | 0.52 ± 0.19 (3) | 102 ± 3 (3) | 28 ± 6 (3) | 74 ± 4 (3) |
| | S310A | 110 ± 16 (3) | 0.22 ± 0.16 (3) | 105 ± 6 (3) | 36 ± 6 (3) | 84 ± 5 (3) |
| Ligand-interacting residues | H94$^{3.29}$F | 139 ± 13 (3) | 13.7 ± 0.4 (3) | 77 ± 11 (3) | N/D | N/D |
| | C97$^{3.32}$A | 142 ± 12 (3) | 0.4 ± 0.3 (3) | 82 ± 13 (3) | 22 ± 10 (3) | 45 ± 7 (3) |
| | R156$^{4.64}$K | 118 ± 13 (3) | 20.7 ± 0.3 (3) | 80 ± 11 (3) | 90 ± 60 (3) | 36 ± 6 (3) |
| | Y237$^{6.51}$A | 108 ± 12 (3) | 109.7 ± 0.2 (3) | 66 ± 5 (3)[a] | 16 ± 8 (3)[b] | 69 ± 5 (3)[b] |
| | I271$^{7.39}$A | 136 ± 14 (3) | 218.8 ± 0.1 (3) | 54 ± 10 (3)[a] | 4 ± 2 (3)[b] | 60 ± 12 (3)[b] |
| Membrane channel | H181$^{5.38}$W | 122 ± 12 (3) | 11.2 ± 0.4 (3) | 91 ± 12 (3) | 8 ± 2 (3) | 75 ± 3 (3) |
| hBLT1 vs. hBLT2 | H94$^{3.29}$Y | 112 ± 7 (3) | 7.33 ± 0.16 (3) | 91 ± 12 (3) | N/D | N/D |
| | G98$^{3.33}$A | 111 ± 11 (3) | 3.7 ± 0.2 (3) | 73 ± 10 (3) | 5.7 ± 1.5 (3) | 84 ± 5 (3) |
| | I271$^{7.39}$T | 122 ± 13 (3) | 126.5 ± 0.3 (3) | 74 ± 12 (3)[a] | 8 ± 3 (3)[b] | 90 ± 20 (3)[b] |
| hBLT1 vs. gpBLT1 | F169$^{ECL2}$L | 112 ± 14 (3) | 3.1 ± 0.2 (3) | 92 ± 6 (3) | 19 ± 3 (3) | 76 ± 5 (3) |
| | P170$^{ECL2}$A | 119 ± 15 (3) | 2.1 ± 0.3 (3) | 94 ± 8 (3) | 20 ± 3 (3) | 61 ± 4 (3) |
| | S264$^{7.32}$R | 110 ± 20 (3) | 1.08 ± 0.14 (3) | 89 ± 2 (3) | 10 ± 2 (3) | 75 ± 4 (3) |
| | N268$^{7.36}$K | 116 ± 13 (3) | 0.59 ± 0.15 (3) | 100 ± 5 (3) | 20 ± 4 (3) | 80 ± 4 (3) |
| | 4 mut | 84 ± 15 (3) | 1.0 ± 0.3 (3) | 101 ± 10 (3) | 5.8 ± 1.2 (3) | 109 ± 6 (3) |
| | gpBLT1-WT | 60 ± 8 (3) | 0.29 ± 0.08 (3) | 107 ± 16 (3) | 132 ± 5 (3) | 58 ± 10 (3) |

Results are expressed as mean ± SEM from at least three independent experiments carried out in triplicate (cell-surface expression data) or quadruplicate (signaling data). The number of independent experiments ($n$) is shown in parenthesis. Cell-surface expression values of mutants are reported as % of hBLT1-WT.
*hBLT1* or *hBLT2* human leukotriene B4 receptor 1 or 2, *gpBLT1* guinea pig BLT1, *IP$_1$* myo-inositol 1 phosphate, *LTB4* leukotriene B4, *WT* wild type, *EC$_{50}$* and *E$_{max}$* potency and efficacy of LTB4, *IC$_{50}$* and *I$_{max}$* potency and efficacy of MK-D-046 inhibition of LTB4-induced IP$_1$ production, *N/D* not determined, *N/A* not available, *CC* crystallization construct, *5 mut* 5 mutations from hBLT1-CC (L106$^{3.41}$W, S116$^{3.51}$Y, A196$^{5.53}$I, C287$^{7.55}$F, and S310A), *ICL3-flav* ICL3-flavodoxin, *Δ 311–352* truncation of hBLT1 residues 311–352, *4 mut* 4 non-conserved residues in the hBLT1 binding pocket mutated to their gpBLT1 equivalents (F169$^{ECL2}$L, P170$^{ECL2}$A, S264$^{7.32}$R, N268$^{7.36}$K).
[a]Maximal efficacy at 1 μM.
[b]Tested with 1 μM of LTB4 as EC$_{80}$ was >1 μM.

small-eared galago (GPCRdb)[29], indicating a distinctive structural feature of hBLT1 that may play an important role in ligand recognition as discussed below.

**Specific interactions with MK-D-046.** MK-D-046 interacts with several residues that have been established to be important for hBLT1 ligand recognition and receptor function. The side chain of R156$^{4.64}$ forms hydrogen bonds with the carbonyl sulfonamide group of MK-D-046. The guanidine group of R156$^{4.64}$ also engages in a stacking interaction with the imidazole ring of H94$^{3.29}$, which, as a 3N-H tautomer, forms hydrogen bonds with both the carbonyl sulfonamide group and the hydroxyl group of the chromanol core of MK-D-046 (Figs. 1c and 3f). The precise polar interaction pattern that H94$^{3.29}$ forms depend upon the tautomerization and orientation of its imidazole ring, which is undistinguishable by electron density. Both H94$^{3.29}$ and R156$^{4.64}$, when individually mutated to alanine by Basu et al.[30], were shown to be critical for LTB4 binding to hBLT1. In our functional and radioligand binding assays, an R156$^{4.64}$K mutation markedly decreased MK-D-046 potency and efficacy (Table 1 and Supplementary Fig. 2b) as well as its ligand binding affinity (Supplementary Table 1 and Supplementary Fig. 3d), likely due to a loss of one or more hydrogen bonds. Mutations of H94$^{3.29}$ either to phenylalanine or to its hBLT2 equivalent residue tyrosine caused a complete loss in MK-D-046 antagonistic activity (Table 1 and Supplementary Fig. 2b, d) as well as an ~870-fold decrease in MK-D-046 binding affinity in the case of H94$^{3.29}$Y (Supplementary Table 1 and Supplementary Fig. 3c). These results indicate that hydrogen bonds with the imidazole ring of H94$^{3.29}$

are critical for binding and that this residue plays a key role in ligand selectivity between hBLT1 and hBLT2.

Among hydrophobic interactions, the largest ligand contact areas are observed for I271$^{7.39}$, C97$^{3.32}$, and F74$^{2.60}$ (Supplementary Table 3), with I271$^{7.39}$ interacting with the chromanol core of MK-D-046, C97$^{3.32}$ with the pyridine ring, and F74$^{2.60}$ with aromatic rings of the benzamide group and the chromanol core (Fig. 1c, d). The effect of an I271$^{7.39}$A mutation on MK-D-046 was difficult to evaluate because it dramatically decreased LTB4 potency (Table 1 and Supplementary Figs. 1b and 2b). On the other hand, C97$^{3.32}$A had little effect on LTB4 but strongly decreased MK-D-046 efficacy, indicating a potential role of this residue in the mechanism of hBLT1 inhibition. Finally, several additional aromatic residues, in particular, F169$^{ECL2}$, W234$^{6.48}$, and F275$^{7.43}$, closely interact with MK-D-046 (Fig. 1c, d and Supplementary Table 3) and contribute to the shape of the MK-D-046 binding site.

**Differences in the ligand-binding site between BLT1 orthologs.** The availability of crystal structures for two different species provides a rare opportunity to analyze differences in the ligand-binding site between BLT1 orthologs. In addition to human and guinea pig, we also considered sequences of mouse and rat BLT1, since both are the most common animals used in preclinical drug testing.

A comparison of the binding pockets of hBLT1 and gpBLT1 structures reveals four non-conserved residues. These residues are four out of five residues that are also not conserved in mouse BLT1 (mBLT1) and rat BLT1 (rBLT1) (Fig. 3a). All four

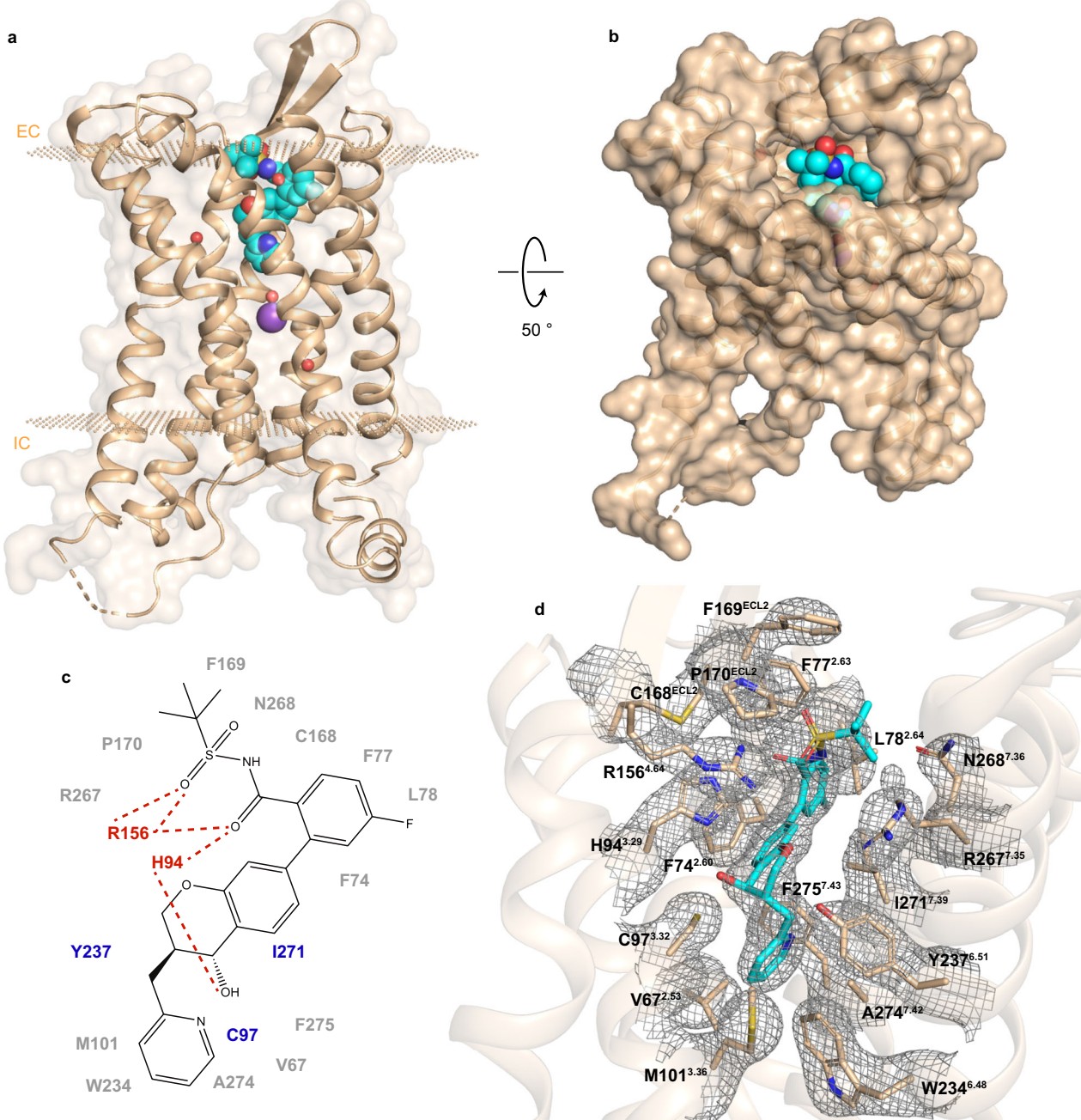

**Fig. 1 Structure and binding site of hBLT1. a** Structure of hBLT1 (wheat cartoon) in complex with MK-D-046 (spheres with cyan carbons). Small spheres (wheat) indicate membrane boundaries (EC, extracellular side; IC, intracellular side), as obtained from the Orientations of Proteins in Membranes (OPM) database[66]. Sodium ($Na^+$, purple) and water (red) are shown as spheres. **b** Extracellular view of hBLT1 showing the solvent-exposed orthosteric binding pocket. **c** Chemical structure of MK-D-046 showing ligand-interacting residues within 4 Å. Critical residues evaluated in our functional and/or binding studies are colored red (polar interactions) or blue (hydrophobic interactions). Hydrogen bonds are shown as dashed red lines. **d** Refined *2mF_o-DF_c* electron density (gray mesh), contoured at 1.0 σ, around MK-D-046 and ligand-interacting residues within 4 Å from MK-D-046.

residues are located on the extracellular side of the binding pocket and include F169ECL2/L171ECL2 (hBLT1/gpBLT1) and P170ECL2/A172ECL2, in addition to S264^7.32/R263^7.32 and N268^7.36/K267^7.36 (Fig. 3f). Out of the four orthologs, hBLT1 is the only one that does not have a large and positively charged amino acid, R or K, at position 7.32 (Fig. 3a). The conformation of R263^7.32 in the gpBLT1 structure (PDB ID 5X33) overlaps with the position of MK-D-046 in the hBLT1 structure (Fig. 3f), indicating that MK-D-046 and other similar ligands may have reduced binding at gpBLT1, as well as mBLT1 and rBLT1. We mutated each of the four non-conserved residues in hBLT1 to their gpBLT1

equivalent and tested them using $IP_1$ production assays. While an S264^7.32R mutation had little effect on MK-D-046 potency, the three other mutations decreased potency by ~2-fold (Table 1 and Supplementary Fig. 2e). The efficacy of MK-D-046 inhibition remained similar for most mutations except for P170ECL2A, which reduced efficacy by ~30%. When combined, all four mutations slightly reduced LTB4 potency, while their effects on MK-D-046 and BIL-260 potencies remained moderate (Table 1 and Supplementary Table 4, Supplementary Figs. 1e and 2e, f). These results are, however, in stark contrast with a slightly increased potency of LTB4, an ~15-fold decreased potency of

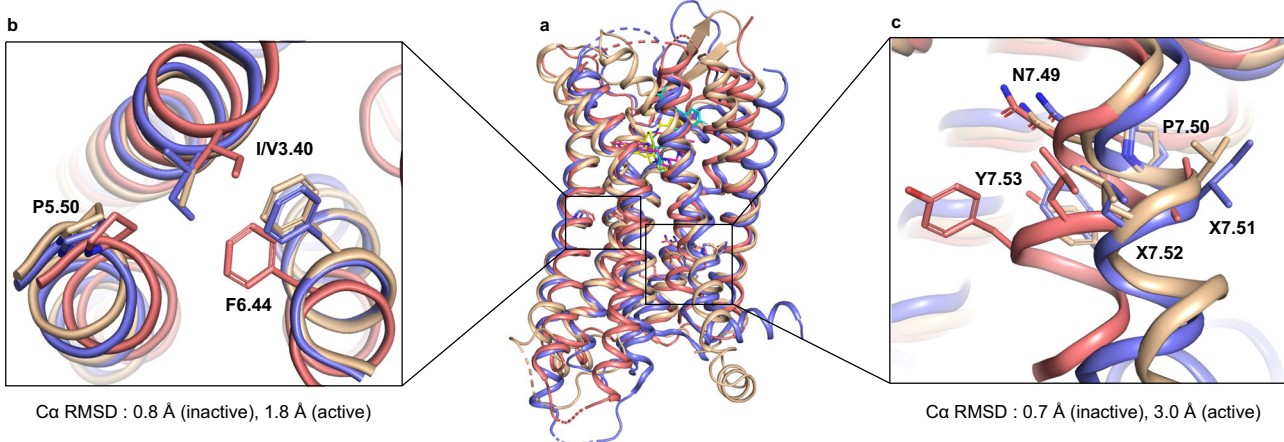

Cα RMSD : 0.8 Å (inactive), 1.8 Å (active)

Cα RMSD : 0.7 Å (inactive), 3.0 Å (active)

**Fig. 2 Comparison of hBLT1 with active and inactive γ branch GPCR structures. a** Overlay of hBLT1 (wheat) with an active κ-opioid receptor (KOR, red, PDB ID 6B73) and inactive KOR (purple, PDB ID 4DJH) structures. **b**, **c** Close-up on P-I/V-F (**b**) and NPxxY motifs (**c**). Root-mean-square deviations of Cα atoms (Cα RMSD) for hBLT1 P-I/V-F and NPxxY motifs with corresponding motifs in active and inactive KOR structures are noted under panels (**b**) and (**c**).

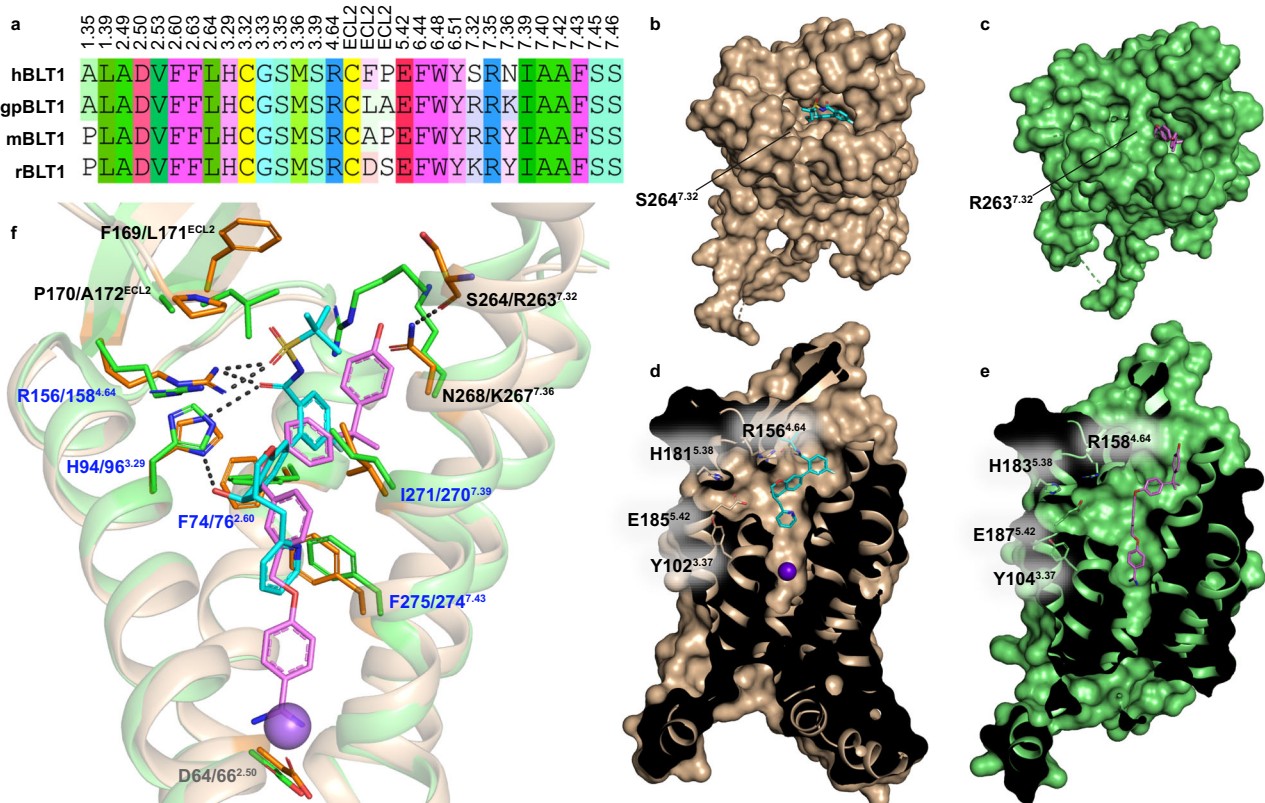

**Fig. 3 Comparison of hBLT1 and gpBLT1 structures. a** Sequence alignment that includes hBLT1 (human) residues within 5 Å from MK-D-046 aligned with gpBLT1 (guinea pig) residues within 5 Å from BIIL-260. These residues are aligned with their equivalent residues in mBLT1 (mouse) and rBLT1 (rat) using ICM-Pro (Molsoft LLC), and the color code represents the property of the amino acid residues (green – hydrophobic, magenta – aromatic, blue – positive charge, red – negative charge, yellow – cysteine, cyan – polar uncharged) as well as the conservation between the two sequences. Residues that are not conserved between the two sequences are shown in different colors. **b**, **c** Extracellular view of hBLT1 (**b**, wheat) and gpBLT1 (**c**, green, PDB ID 5X33) structures showing differences in the access to the binding site with co-crystallized ligands MK-D-046 (cyan) and BIIL-260 (magenta). Locations of non-conserved residues at position 7.32 are noted. **d**, **e** Cross-section of hBLT1 (**d**, wheat) and gpBLT1 (**e**, green) binding pockets showing the membrane channel on the left, with residues $Y^{3.37}$, $H^{5.38}$, and $E^{5.42}$ that shape the channel relative to $R^{4.64}$. **f** Overlay of hBLT1 (wheat) and gpBLT1 (green) binding pockets, with residues that differ between structures represented as sticks. Conserved residues that differ in their conformation are labeled in blue, non-conserved residues are black, and $D^{2.50}$ is gray for reference of the sodium ($Na^+$) site location. Residues are noted in hBLT1/gpBLT1 order. Hydrogen bonds are represented as dashed lines. $Na^+$ of hBLT1 is shown as a violet sphere in **d** and **f**.

MK-D-046, and an ~4-fold decreased potency of BIIL-260 at gpBLT1-WT compared to hBLT1-WT, suggesting that differences in residues outside the ligand-binding pocket may play a larger role in commonly observed variations of ligand potencies between species. For example, the potency rank order of compounds related to LTB4 was shown to be reversed between hBLT1 and gpBLT1[31]. In another study, several BLT1 antagonists demonstrated different affinities between receptors from human, rat, guinea pig, and dog species[32]. These differences in receptor structure and ligand-binding properties between orthologs should be taken into account when conducting drug testing in different species.

Apart from the four non-conserved residues that vary between hBLT1 and gpBLT1, there are several other residues in the ligand-binding pocket that, although conserved, differ in their conformation between structures. These variations are mainly due to the interactions of these residues with the chemically distinct co-crystallized ligands but may also result from an overall difference between the structures of these two orthologs. As previously mentioned, both H94[3.29] and R156[4.64] in hBLT1 form hydrogen bonds with MK-D-046 (Figs. 1c and 3f). In gpBLT1, R158[4.64] does not interact with BIIL-260 (Fig. 3f), and its guanidine group is shifted ~3.0 Å from its equivalent position in the hBLT1 structure. The imidazole ring of H94[3.29] is rotated by ~30° compared to its orientation in gpBLT1. Other ligand-interacting residues such as I271[7.39], F74[2.60], and F275[7.43] have 1.0–1.5 Å shifts or differences in rotation from their gpBLT1 equivalent residues. These differences in residue conformations slightly alter the shape of the binding pocket between hBLT1 and gpBLT1 structures.

An additional variation between the hBLT1 and gpBLT1 binding pockets is found in the conserved sodium ($Na^+$) binding site[33]. In the gpBLT1 structure (PDB ID 5X33), the benzamidine group of BIIL-260 reaches deep into the binding pocket and interacts with the sodium site residues[23]. Although MK-D-046 does not extend down to the sodium site in hBLT1 (Fig. 3f), the sodium site in hBLT1 retains a conformation similar to that of gpBLT1, with a clear electron density inside the site (Supplementary Fig. 4h). While both sodium and benzamidine were present in the crystallization conditions for hBLT1, we modeled a $Na^+$ in this site because benzamidine would clash with MK-D-046 (Fig. 3f).

**Putative channel for accessing the ligand-binding site in BLT1 directly from the membrane**. Structures of many lipids, as well as several non-lipid GPCRs, revealed the existence of channels buried in the lipid bilayer that allow ligands to access their respective orthosteric binding pockets directly from the membrane[34,35]. Similarly, the shape of the ligand-binding pocket in the hBLT1 structure suggests the existence of a channel extending directly into the lipid membrane between TM4 and TM5 (Fig. 3d and Supplementary Fig. 7a, b) that may serve as an access route for LTB4, which is a lipid chemoattractant. An almost identical membrane channel exists in the gpBLT1 structure (PDB ID 5X33) (Fig. 3e), except with a slightly smaller opening, which may be due to the overall difference between structures and the lower resolution of the gpBLT1 structure. hBLT1 residues that shape this channel, Y102[3.37], H181[5.38], and E185[5.42] (Fig. 3d and Supplementary Fig. 7b–d), have been identified to affect LTB4 signaling here (Table 1 and Supplementary Fig. 1c) and in previous studies[30,36], providing additional support that LTB4 can use this channel to enter the pocket.

**Molecular determinants of antagonist recognition at hBLT1**. To gain additional insights into ligand specificity and selectivity,

we performed molecular docking of a panel of antagonists, including preclinical and clinical drug candidates. We started with assessing the performance of the current hBLT1 and the previously solved gpBLT1 crystal structures in docking of the co-crystallized ligands. The very different chemical scaffolds of MK-D-046 and BIIL-260 provided an opportunity for a direct cross-docking evaluation. Our hBLT1 structure allowed for reproducible docking poses of both MK-D-046 and BIIL-260 that were consistent with their poses in the crystal structures (RMSD values 0.9 Å and 1.6 Å for hBLT1 and gpBLT1, respectively). However, our attempt to dock MK-D-046 in the gpBLT1 crystal structure (PDB ID 5X33) resulted in unsatisfactory docking scores and poorly reproducible poses that were inconsistent with the crystal structure (best RMSD = 3.8 Å), apparently, due to the overlap with the R263[7.32] side-chain conformation, as discussed above. Although the docking pose of BIIL-260 in gpBLT1 was consistent with the crystal structure (RMSD = 1.2 Å), the docking score was worse than with hBLT1 (Supplementary Table 5). Moreover, docking of several other clinical and preclinical compounds strongly preferred the hBLT1 over the gpBLT1 crystal structure. The difference in docking results is likely due to the differences in their binding pockets, outlined in the previous section, as well as the differences in the accuracy of atomic coordinates of these two structures due to their different resolutions. Our docking results, therefore, revealed substantial differences between the two crystal structures and suggested the better utility of the hBLT1 structure in structure-based ligand discovery.

After validating our approach, we proceeded with using the hBLT1 structure to evaluate docking of several MK-D-046 analogs developed for the treatment of T2D and other inflammatory conditions[24] as part of structure-activity relationship (SAR) studies (Fig. 4 and Supplementary Table 6). These analogs differ from MK-D-046 by their substituents ($R_1$–$R_4$) around the chromanol core, which influences their potency (Fig. 4 and Supplementary Table 6)[24]. A bulky *tert*-butyl in position $R_1$ helps to anchor the carbonyl sulfonamide group in an optimal orientation for polar interactions with H94[3.29] and R156[4.64]. Substituting the *tert*-butyl group with a slightly smaller cyclopropane (Example 16) reduces the ligand potency by 2-fold compared to MK-D-046, whereas changing it to an even smaller methyl group (Example 4) results in an ~50-fold decrease in potency (Fig. 4b, e and Supplementary Table 6). As a result of a smaller substituent, the carbonyl sulfonamide group of Example 4 rotates, leading to less optimal hydrogen bonds with R156[4.64] (Fig. 4b). A fluorine in position $R_2$ of MK-D-046 binds in a hydrophobic sub-pocket containing F74[2.60] and L78[2.64]. The removal of this fluorine (Example 12) causes an ~10-fold decrease in potency, although the docking score remains unchanged (Fig. 4c and Supplementary Table 6). The size of the substituent in position $R_4$ on the other side of the chromanol core near the bottom of the binding pocket is also important. Substituting the pyridine ring with a larger ethyl-diazole group (Example 37), which could clash with F275[7.43], results in an ~2-fold decrease in potency compared to Example 12 (Fig. 4f and Supplementary Table 6). Finally, changing chirality of the $R_3$ and $R_4$ substituents (Example 13) leads to the largest decrease in potency by ~220-fold, which is likely due to a shift of the chromanol core in order to maintain a hydrogen bond between the hydroxyl group and H94[3.29] (Fig. 4d and Supplementary Table 6). Overall, these results provide additional support for H94[3.29] and R156[4.64] as important molecular determinants of ligand recognition, since maintaining optimal hydrogen bonds with both residues is important for sustaining a high potency. In addition, the residues that line the bottom of the binding pocket, which include F275[7.43], do not accommodate large substituents as seen with Example 37.

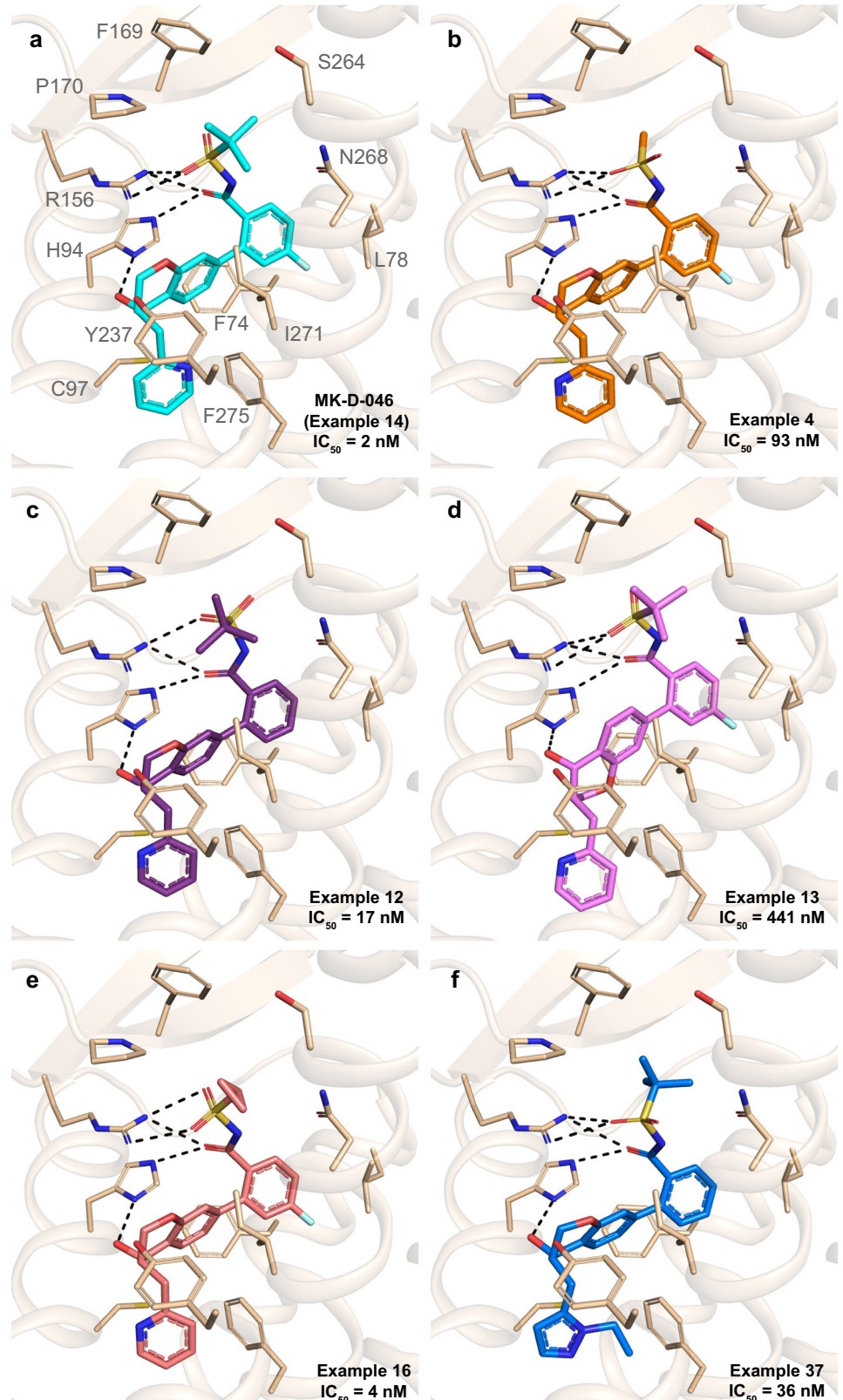

**Fig. 4 SAR and docking studies for MK-D-046 and its analogs. a** Binding mode of MK-D-046 in hBLT1 as revealed in the crystal structure. **b-f** Binding modes of Examples 4, 12, 13, 16, and 37 from ref. [24] obtained by docking in the hBLT1 structure. Inhibition potencies (IC$_{50}$) for all ligands, obtained using LTB4-induced cAMP assays in hBLT1-expressing HEK293 cells, were taken from ref. [24]. Hydrogen bonds are shown as dashed lines.

**Docking provides insights into LTB4 recognition at hBLT1.** It has been widely accepted that most agonists do not simply lock a receptor in a fully active conformation but rather shift the dynamic equilibrium between multiple states. Thus, we attempted docking of LTB4 in the inactive hBLT1 structure with an expectation that it may shed some light on the binding of endogenous ligands. Docking of LTB4 in the crystal structure of hBLT1 revealed two possible overall conformations: conformation 1, where the tail of LTB4 occupies the putative membrane channel, and conformation 2, where LTB4 extends deeper into the orthosteric binding pocket towards the sodium site (Supplementary Figs. 7c, d and 8). Both conformations can accommodate either the most stable all-trans conformer of the triene group (conformations 1a and 2a) or a distorted cis-trans conformer (1b and 2b), which has been proposed in previous studies[37] (Supplementary Fig. 8). The H181$^{5.38}$W mutation, which was designed to block the entrance of the membrane channel, decreased LTB4 potency by ~20-fold with no effect on MK-D-046 potency (Table 1 and Supplementary Figs. 1c, 2c). Radioligand binding assays confirmed the absence of direct interactions between the ligands and this residue (Supplementary Table 1, and Supplementary Fig. 3f). While our results and those from previous studies[30,36] support conformation 1 of LTB4 (Supplementary Figs. 7c and 8a, b), implicating the prospect of a membrane channel for BLT1, further studies are required to establish the mode of entrance of LTB4 and its binding pose.

In all four predicted LTB4 conformations, the carboxyl group of LTB4 is anchored by specific polar interactions with R156$^{4.64}$, and in conformations 1a and 2a, the carboxyl group also interacts with H94$^{3.29}$ (Supplementary Fig. 8). In addition, the 5-hydroxyl group of LTB4 is predicted to make a hydrogen bond with H94$^{3.29}$ (conformation 1b) or with N268$^{7.36}$ (conformation 2a). Accordingly, mutations of two of these residues, H94$^{3.29}$F and R156$^{4.64}$K, decreased LTB4 potency by ~28- and 40-fold, respectively (Table 1 and Supplementary Fig. 1b). In binding assays, R156$^{4.64}$K caused an ~3-fold decrease in LTB4 affinity (Supplementary Table 1 and Supplementary Fig. 3d). In all conformations except 1a, the 12-hydroxyl group of LTB4, which was shown to be important for high-affinity binding[36], is predicted to occupy a similar position as the hydroxyl group of the co-crystallized MK-D-046 compound and form a hydrogen bond with H94$^{3.29}$ (Supplementary Fig. 8). In conformation 1a, the 12-hydroxyl group is predicted to form a hydrogen bond with the side chain of Y237$^{6.51}$ and/or with the side chain of E185$^{5.42}$ through a water molecule or directly upon possible tightening of the binding pocket during receptor activation. Mutating H94$^{3.29}$ to its hBLT2 equivalent residue tyrosine resulted in an ~14-fold decrease in LTB4 potency, indicating that formation of a hydrogen bond with H94$^{3.29}$ may contribute to the subtype selectivity of LTB4 (Table 1 and Supplementary Fig. 1d).

The phenol ring of Y237$^{6.51}$ is predicted to form π-π interactions with the 14-en of LTB4 (Supplementary Fig. 8), which explains our observation that a Y237$^{6.51}$A mutation (Table 1 and Supplementary Fig. 1b) causes an ~200-fold decrease in LTB4 potency and a >30% decrease in efficacy, while the previous studies[36] showed no effect of a Y237$^{6.51}$F mutation on LTB4 binding. Also due to their close proximity (~3.7 Å), Y237$^{6.51}$ may be involved in π-π interactions with W234$^{6.48}$, which was shown to be an important residue for LTB4 binding[37].

The rigid triene group of LTB4 is predicted to interact with I271$^{7.39}$ in all conformers (Supplementary Fig. 8). Mutations of I271$^{7.39}$ to alanine or to its hBLT2 equivalent residue, threonine, caused an ~400-fold or 250-fold decrease in LTB4 potency, respectively (Table 1 and Supplementary Fig. 1b,d). In binding assays, an I271$^{7.39}$T mutation resulted in a >20-fold decrease in LTB4 affinity (Supplementary Table 1 and Supplementary Fig. 3e).

This supports the importance of hydrophobic interactions between this residue and LTB4 (Supplementary Fig. 8), which may also contribute to the selectivity of LTB4 towards hBLT1 versus hBLT2. Another non-conserved residue between the two subtypes in the binding pocket, G98$^{3.33}$, was mutated to its hBLT2 equivalent residue, alanine. This mutation decreased both LTB4 potency (~7-fold) and efficacy (~30%) (Table 1 and Supplementary Fig. 1d), suggesting a difference in the mechanism of activation between hBLT1 and hBLT2.

Further, we tested reverse mutations of two hBLT2 residues to hBLT1 equivalents, Y98$^{3.29}$H and T274$^{7.39}$I, in IP$_1$ production assays with BLT2-selective agonist 12-hydroxyheptadecatrienoic acid (12(S)-HHTrE)[7] and BLT2-selective antagonist LY-255283[8] (Supplementary Table 7 and Supplementary Fig. 9). The first mutation, Y98$^{3.29}$H, had little effect on both agonist and antagonist, suggesting that this residue does not interact with BLT2-selective ligands. The second mutation decreased antagonist potency by ~5-fold, indicating that T274$^{7.39}$ contributes to the BLT2 selectivity of antagonists.

## Discussion

Many studies have shown profound effects of BLT1 on numerous inflammatory processes and established its link to inflammatory diseases. Due to its connection to inflammation and its cell-surface location as a GPCR, various BLT1 ligands have been designed and tested in clinical trials[38] for a range of inflammatory diseases including asthma, COPD, rheumatoid arthritis, inflammatory bowel disease, and cancer[18,19]. Here we presented a crystal structure of hBLT1 co-crystallized with antagonist MK-D-046, which was part of a panel of BLT1 antagonists developed for the treatment of T2D and other conditions that may be responsive to BLT1 antagonism[24]. Several studies have shown strong support for BLT1 as a potential target for the therapeutic treatment of T2D and related health conditions. One study demonstrated that BLT1 deficiency improves glucose and insulin tolerance in a diet-induced obese mouse model[10]. In another study, BLT1 antagonism was shown to reduce inflammation and insulin resistance in both diet-induced and genetic-mediated obese mouse models[39]. Over 460 million adults worldwide, aged 20–79, (~9% of this age group) have diabetes, and ~4.2 million deaths from the same age group are associated with diabetes[40]. By the year 2045, it is estimated that there will be ~700 million adults from the same age group (over 50% increase) living with diabetes in the world. Unfortunately, many current treatments for diabetes have significant adverse effects, such as hypoglycemia, gastrointestinal side effects, increased cholesterol, and heart failure[41]. Targeting T2D, which makes up ~90% of diabetes cases in adults[42], through BLT1 may provide better alternatives for diabetes therapeutics.

The crystal structure of hBLT1, combined with extensive site-directed mutagenesis and docking studies, provides insights into hBLT1 ligand recognition, its mechanism of action, as well as subtype and species selectivity. Here we identified H94$^{3.29}$, R156$^{4.64}$, Y237$^{6.51}$, and I271$^{7.39}$ as critical residues for hBLT1 ligand recognition, since these residues significantly affect LTB4 and MK-D-046 potency, efficacy, and affinity. Docking studies confirmed both H94$^{3.29}$ and R156$^{4.64}$ as key residues for maintaining important interactions with MK-D-046 and its analogs. Site-directed mutagenesis of hBLT1 and hBLT2 showed that residues 3.29 and 7.39 impact ligand recognition and subtype selectivity. We also demonstrated that Y237$^{6.51}$ is important for LTB4 potency and efficacy through its π–π interactions with LTB4 in hBLT1. A recent study by Kim et al.[43] found that the equivalent residue Y240$^{6.51}$ in hBLT2 also impacts LTB4 potency and efficacy, however, it acts via forming a hydrogen bond to the

12-hydroxyl group of LTB4 rather than through π–π interactions. The same study[43] showed that LTB4 binds to hBLT2 in a U-shape conformation, which is similar to a LTB4 binding pose in hBLT2 previously obtained by NMR[44]. This differs considerably from the docking models of LTB4 bound in hBLT1 presented here. Based on this information, it is possible that hBLT1 and hBLT2 have different binding modes for LTB4, which might contribute to different affinities and mechanisms of activation between receptors. Our structural and site-directed mutagenesis studies may provide aid in the design of more selective ligands, which can improve our understanding of the distinct signaling pathways of BLT1 and BLT2, since both receptors not only couple to different G proteins but also have contrasting functions. A study by Zinn et al.[45] found that BLT1 and BLT2 have opposing roles in the sensitization of peripheral sensory neurons. BLT1 and BLT2 also have divergent and opposite effects in inflammatory diseases such as asthma[15,46] and cancer[17].

Our hBLT1 structure also provides key advantages in docking and critical insights in species selectivity. We identified structural differences between hBLT1 and gpBLT1, which are mainly located within their orthosteric binding pockets, as the result of differences in sequence, resolution, and chemically distinct ligands. Both co-crystallized ligands, MK-D-046 and BIIL-260, have lower potency and inhibition efficacy in gpBLT1 than hBLT1. This difference is due in part to several non-conserved residues (hBLT1/gpBLT1: $F169^{ECL2}/L171^{ECL2}$, $P170^{ECL2}/A172^{ECL2}$, $S264^{7.32}/R263^{7.32}$, and $N268^{7.36}/K267^{7.36}$) near the extracellular side of the binding pockets, but mostly because of non-conserved residues outside the binding pocket. Even though many residues within the orthosteric binding pockets of hBLT1 and gpBLT1 are conserved, several residues important for ligand recognition, such as $H94/H96^{3.29}$, $R156/R158^{4.64}$, and $I271/270^{7.39}$, vary in conformation between hBLT1 and gpBLT1 structures. The differences in conformation are likely due to the specific interactions with the chemically distinct co-crystallized ligands of hBLT1 and gpBLT1 structures, which also contribute to the differences in docking results, where the hBLT1 structure proved to be more suitable for docking than the gpBLT1 structure. Therefore, the hBLT1 structure serves as a better template for the future design of hBLT1 ligands and reveals the importance of human structures.

Although our hBLT1 structure is in the inactive state, docking and functional studies gave additional insights into agonist LTB4 binding. The LTB4 recognition data presented here may lead to better therapeutics for health conditions where BLT1 plays a protective role, which includes viral infection[16], lipopolysaccharide-induced acute lung injury[47], and cancer[17]. Although functional and docking studies supported the presence of a membrane channel, additional experiments are needed to verify whether LTB4 enters the pocket directly from the membrane through this channel or through the extracellular opening.

Overall, the hBLT1 structure in complex with MK-D-046 offers a deeper understanding and clarification of hBLT1 ligand recognition that was not fully achieved using the gpBLT1 structure. Going forward, the hBLT1 structure presented here will serve as a foundation for hBLT1 ligand recognition, hBLT1 species and subtype selectivity, and the development of more effective therapeutics for inflammatory diseases.

## Methods

**Generation of the hBLT1 crystallization construct.** The vector containing the hBLT1-WT (UniProt ID Q15722) sequence with N-terminal (Δ 1–4) and C-terminal (Δ 311–352) truncations was received as a gift from iHuman Institute (ShanghaiTech University, China). The nucleotide sequence of truncated hBLT1 (Δ 1–4 and Δ 311–352) was codon-optimized for insect cell expression (GenScript) and cloned in a modified pFastBac1 (Invitrogen) baculovirus expression vector. At the N terminus of hBLT1, the plasmid contained a hemagglutinin signal sequence (KTIIALSYIFCLVFA), FLAG tag, and AscI site. At the C terminus was an EcoRI site, a PSP recognition site, and a 10×His tag. Site-directed mutagenesis was carried out by using AccuPrime Pfx DNA polymerase (ThermoFisher Scientific) and oligonucleotides (Integrated DNA Technologies) with internal mismatches (Supplementary Table 8). All sequences were verified by Sanger sequencing (Genewiz).

Several additional modifications to the initial truncated hBLT1 sequence (Δ 1–4, Δ 311–352) were made to achieve the crystallization construct (hBLT1-CC). The protein flavodoxin (PDB ID 1I1O, with mutation Y98W) was fused in the ICL3 of hBLT1 between residues R212 and F213. Two arginine residues were added as a linker between the last residue of flavodoxin (isoleucine) and hBLT1 residue F213. Mutations $L106^{3.41}W$, $S116^{3.51}Y$, $A196^{5.53}I$, $C287^{7.55}F$, and S310A were introduced to improve protein stability.

Three mutations ($L106^{3.41}W$, $S116^{3.51}Y$, and S310A) were obtained from different literature sources: $S^{3.51}Y$ is part of the conserved DRY motif in class A GPCRs[26,27], $L^{3.41}W$ was found to improve the stability of β$_2$AR, which is another class A receptor[48], and S310A is a phosphorylation site of hBLT1[49] and is equivalent to the stabilizing mutation found in the gpBLT1 structure (PDB ID 5X33)[23]. Two additional thermostabilizing mutations ($A196^{5.53}I$ and $C287^{7.55}F$) were selected from screening a set of 48 mutations, predicted by CompoMug[50] v. 0.1, for yield, monodispersity, and thermostability.

**Expression and purification of the hBLT1 crystallization construct.** hBLT1-CC was expressed in *Spodoptera frugiperda* (Sf9, ATCC, CRL-1711, authenticated by supplier using morphology and growth characteristics, certified mycoplasma-free) insect cells using the Bac-to-Bac Baculovirus Expression System (ThermoFisher Scientific). Cells at a density of $(1–3) × 10^6$ cells mL$^{-1}$ were infected with baculovirus at 27 °C using a multiplicity of infection of 5. Cells were collected by centrifugation 48 h after infection and stored at −80 °C until use. Expression was done in separate 1 L volumes of biomass.

Frozen biomass was thawed and resuspended in hypotonic buffer (10 mM HEPES pH 7.5, 10 mM MgCl$_2$, 20 mM KCl, and in-house protease inhibitor cocktail). Membrane fractions were isolated from 1 L or 2 L of biomass by repeated Dounce homogenization and ultracentrifugation for 25 min at 4 °C and 167,000×g in hypotonic buffer (twice) and hypertonic buffer (twice; 10 mM HEPES pH 7.5, 10 mM MgCl$_2$, 20 mM KCl, 1 M NaCl, and in-house protease inhibitor cocktail).

Washed membranes were incubated at 4 °C for 1 h in a hypotonic buffer in the presence of 2 mg mL$^{-1}$ iodoacetamide (Sigma-Aldrich), 100 μM MK-D-046 (synthesized as described[24]), and in-house protease inhibitor cocktail. Receptor was subsequently extracted from membranes in a volume of 50 mL (for 1 L biomass) or 100 mL (for 2 L biomass) by the addition of 2× solubilization buffer (100 mM HEPES pH 7.5, 1600 mM NaCl, 10% (v/v) glycerol, 2% (w/v) n-dodecyl-β-D-maltoside (DDM, Anatrace), and 0.4% (w/v) cholesterol hemisuccinate (CHS, Sigma-Aldrich)) for 2.5 h at 4 °C. After high-speed ultracentrifugation for 1 h at 4 °C and 371,000×g, the supernatant was incubated overnight at 4 °C in the presence of 1 mL of TALON (immobilized metal affinity chromatography, IMAC) resin (Takara) and 20 mM imidazole pH 7.5.

Following overnight incubation, the sample was washed on a gravity column (Bio-Rad) with 10 column volumes (cv) of wash buffer 1 (100 mM HEPES pH 7.5, 800 mM NaCl, 20 mM MgCl$_2$, 20 mM imidazole pH 7.5, 10% (v/v) glycerol, 0.05%/0.01% (w/v) DDM/CHS, 8 mM ATP (Sigma-Aldrich, buffered in 50 mM HEPES pH 7.5), and 50 μM MK-D-046) followed by 9 cv of wash buffer 2 (100 mM HEPES pH 7.5, 150 mM NaCl, 20 mM imidazole pH 7.5, 10% (v/v) glycerol, 0.05%/0.01% (w/v) DDM/CHS, and 50 μM MK-D-046). The sample was eluted using 4 cv of elution buffer (100 mM HEPES pH 7.5, 150 mM NaCl, 200 mM imidazole pH 7.5, 10% (v/v) glycerol, 0.025%/0.005% (w/v) DDM/CHS, and 50 μM MK-D-046); however, the first 0.5 cv of elution buffer flow-through was discarded. The sample was subsequently concentrated to 300–400 μL using an Amicon Ultra – 4 Centrifugal filter with 100 kDa molecular weight cutoff (Millipore). The concentrated sample, 30 IU of His-tagged PreScission Protease (GenScript), and 20 UN of His-tagged PNGaseF (Sigma-Aldrich) were concomitantly passed over a PD MiniTrap G-25 desalting column (GE Healthcare) to remove imidazole and adjust detergent concentration to 0.05%/0.01% (w/v) DDM/CHS. After overnight incubation at 4 °C, cleaved tags and proteases were removed with reverse IMAC by binding to 80 μL of TALON resin for 3.5 h at 4 °C. After the sample had flown through reverse IMAC, the resin was washed with 2.75 cv of reverse IMAC buffer (100 mM HEPES pH 7.5, 150 mM NaCl, 10% (v/v) glycerol, 0.015%/0.003% (w/v) DDM/CHS, and 50 μM MK-D-046) to remove any additional sample. The receptor was then concentrated to 30–40 mg mL$^{-1}$ using an Amicon Ultra – 0.5 mL Centrifugal Filter with 100 kDa molecular weight cutoff (Millipore). A solution of MK-D-046 (10 mM in DMSO) was added to the final concentrated protein sample so that the final concentration of DMSO was 5% (v/v). (The solution of 10 mM MK-D-046 was made by dissolving MK-D-046 in DMSO, which was used as a stock solution for all instances where MK-D-046 was used in purification).

Over 100 hBLT1 constructs were designed, cloned, expressed, purified, characterized, and optimized for structural studies before achieving the hBLT1-CC construct. The most stable constructs were each screened with a panel of 20 ligands before selecting MK-D-046 for crystallization.

**Crystallization in LCP.** Purified hBLT1-CC in complex with MK-D-046 was reconstituted into LCP by mixing two volumes of purified and concentrated

receptor solution with three volumes of molten monoolein (Sigma-Aldrich)/cholesterol (Sigma-Aldrich) (9:1 w/w) using a mechanical syringe mixer[51]. LCP crystallization trials were performed in 96-well glass sandwich plates (Marienfeld) using an NT8-LCP crystallization robot (Formulatrix) by dispensing 40 nL of protein-laden LCP and 800 nL of precipitant solution per well.

hBLT1-CC crystallized in a range of conditions. After optimizing for size and diffraction quality, the final crystals used for data collection grew in 100 mM sodium citrate tribasic dihydrate pH 5.8, 385–500 mM sodium acetate trihydrate, 6–32 mM benzamidine hydrochloride, 27–30% (v/v) PEG-400, 10 μM MK-D-046, and 1% (v/v) DMSO (DMSO is the solvent for MK-D-046) (Supplementary Fig. 4b, c). Crystals began to appear 24–48 h after incubation at 20 °C and continued to grow for up to 2 weeks. An average crystal size was $40 \times 15 \times 8$ μm. Crystals were harvested from LCP using 30–100 μm micromounts (MiTeGen) and flash-frozen in liquid nitrogen for data collection.

**Crystallographic data collection, structure solution, and structure refinement.** X-ray diffraction data were collected remotely using the data acquisition software JBluIce v. 2018.2 Build 6401 at the GM/CA beamline 23ID-B, equipped with an Eiger-16M detector (Dectris), at the Advanced Photon Source (APS), Argonne National Laboratory, IL, USA. The crystals were exposed with a 10 μm X-ray minibeam with the wavelength of 1.0332 Å (energy 12 keV), for 0.5–1.0 s, 0.2° oscillation per frame, and 11–150 frames per crystal. HKL-2000 v.718.05[52] was used for indexing, integrating, scaling, and merging data from 32 crystals of hBLT1-CC in complex with MK-D-046. The final dataset was analyzed and anisotropically truncated at 2.9 Å (a*) × 2.9 Å (b*) × 3.6 Å (c*) resolution by the STARANISO server v. 2.6.32[53].

Initial phase information of hBLT1-CC in complex with MK-D-046 was obtained by molecular replacement (MR) with Phaser MR[54] (ccp4 v. 7.0.078) using gpBLT1[23] (PDB ID 5X33 with removed ligand, T4L, and residues 214–220), and flavodoxin[55] (PDB ID 1I1O) as search models. The correct MR solution contained one molecule per asymmetric unit of the P $2_1$ $2$ $2_1$ lattice. The structure was improved iteratively through cycles of automatic refinement with Phenix v. 1.17.1-3660[56] or Buster v. 2.10.2[57] followed by manual examination and rebuilding of the refined coordinates in the program COOT v. 0.8.9[58] using both $2mF_o$-$DF_c$ and $mF_o$-$DF_c$ maps. The final refinement was performed with Phenix. Ligand restraints were generated using Phenix eLBOW[59]. The final model of the hBLT1 structure in complex with MK-D-046 contains 298 residues of hBLT1 (residues 13–310), 147 residues of flavodoxin (residues 1002–1148), two arginine residues at the ICL3 junction site (residues 1149 and 1150), an *Eco*RI site (residues 311–312), and part of a PSP recognition site (313–318). Data collection and refinement statistics of the hBLT1 structure are summarized in Supplementary Table 2. All structural images in this manuscript were generated with PyMOL v.2.3.3 (Schrödinger).

We observed that MK-D-046 can be placed in the binding pocket in two possible conformations that differ by an ~180° flip of the chromanol core and pyridine ring (Supplementary Fig. 4f, g). The conformation in Supplementary Fig. 4f was chosen for the final hBLT1 structure because it had an overall better fit in the density and an absence of strong $mF_o$-$DF_c$ densities at ±3.0 σ. The conformation in Supplementary Fig. 4g could form a hydrogen bond with Y237[6.51]. However, a Y237[6.51]A mutation had little effect on MK-D-046 potency (Table 1 and Supplementary Fig. 2b), indicating that this residue is unlikely to form any significant interactions, such as a hydrogen bond, with MK-D-046.

Before achieving the final hBLT1 crystal structure, over 260 crystal samples were prepared for data collection and evaluated by X-ray diffraction. Prior to data collection, over 350 crystal samples were prepared and screened for diffraction quality. Overall, more than 190 optimization plates were designed and utilized to achieve the final crystallization conditions for hBLT1-CC. Over 400 rounds of data processing and over 600 cycles of structure refinement were carried out to obtain the final hBLT1 crystal structure.

**Plasmids for IP₁ production assays.** The hBLT1-WT sequence (UniProt ID Q15722) cloned into a pcDNA3.1+ (Invitrogen) vector at *Eco*RI (5′) and *Xho*I (3′) was purchased from cDNA.org. The hBLT1-CC, 5 mut, ICL3-flav, hBLT2-WT (UniProt ID Q9NPC1), and gpBLT1-WT (UniProt ID Q9WTK1) sequences were synthesized by GenScript. All plasmids were synthesized with or modified to include a 3× hemagglutinin (HA) tag (YPYDVPDYA) at the N terminus. All modifications (point mutations, truncations, and insertions) were made and verified similar to hBLT1-CC used in insect cell expression. All oligonucleotides used for site-directed mutagenesis are listed in Supplementary Table 8.

**Cell-surface expression of receptor constructs used for IP₁ production assays.** HEK293 cells (ATCC, CRL-1573, used between passages 5–25) were seeded in 24-well plates precoated with 0.1 mg mL⁻¹ poly-L-lysine (Sigma-Aldrich) at 70,000 cells per well and transfected at the same time with 500 ng of plasmid coding for 3×HA-tagged hBLT1-WT, gpBLT1-WT, hBLT2-WT, or mutants using Lipofectamine 3000 (Invitrogen) as previously described[60]. At 48 h post-transfection, cells were fixed with 3.7% (w/v) formaldehyde in Tris-buffered saline (TBS, 20 mM Tris-HCl pH 7.5, and 150 mM NaCl) for 5 min at RT. Cells were washed three times with TBS and incubated for 1 h in TBS supplemented with 3% fat-free milk (w/v) in order to block non-specific binding sites. A rat monoclonal anti-HA

antibody conjugated to peroxidase (clone 3F10, Roche, cat# 12 013 819 001) was added at 1:1000 dilution in TBS-3% milk for 3 h at RT. Following incubation, cells were washed three times with TBS before the addition of 250 μL of 3,3′,5,5′-tetramethylbenzidine (TMB, Sigma-Aldrich). Plates were incubated at RT for 5 to 15 min and the reaction was stopped by the addition of 250 μL of HCl 2 N. 100 μL of the yellow reaction was transferred into a 96-well plate and the absorbance was read at 450 nm on a Mithras2 LB943 multimode microplate reader (Berthold Technologies). Cells transfected with an empty pcDNA3.1+ vector were used to determine the background.

**IP₁ production assays.** The IP-One kit (Cisbio, 62IPAPEB) was used according to the manufacturer's instructions. HEK293 cells were seeded onto poly-L-lysine-coated 384-well plates at 20,000 cells per well and transfected with 10 ng of DNA coding for the human Gα₁₆ protein[61] (for hBLT1 and gpBLT1) or Gα₁₄ protein[7] (for hBLT2) and 30 ng of DNA coding for 3×HA-tagged hBLT1-WT, gpBLT1-WT, hBLT2-WT, or mutants using the Lipofectamine 3000 reagent based on previously published protocols[62,63]. At 48 h post-transfection, the media was removed, and the cells were washed with PBS. For the agonist mode assay, cells were stimulated with increasing concentrations of LTB4 (Cayman Chemical, for hBLT1 and gpBLT1) or 12(S)-HHTrE (Cayman Chemical, for hBLT2) in stimulation buffer. For antagonist testing, cells were stimulated directly with a concentration of ligand (LTB4 or 12(S)-HHTrE) corresponding to the EC₈₀ of the WT or mutant receptor and with increasing concentrations of the antagonists MK-D-046, BIIL-260 (Sigma-Aldrich), or LY-255283 (Tocris) in IP₁ stimulation buffer. After 1 h stimulation at 37 °C, the cells were lysed with IP₁-D2 and Ab-Crypt reagents in Lysis Buffer and then incubated at RT for at least 1 h. The plate was read on a Tecan GENios Pro multimode plate reader using an HTRF filter set ($\lambda_{ex}$ 320 nm, $\lambda_{em}$ 620 and 655 nm). Data were analyzed and plotted using Prism v. 9 (GraphPad, San Diego, CA).

**Membrane preparation for binding assays.** The vector containing the hBLT1-WT (UniProt ID Q15722) sequence was received as a gift from iHuman Institute (ShanghaiTech University, China). The nucleotide sequence of hBLT1-WT was codon-optimized for insect cell expression (GenScript) and cloned in a modified pFastBac1 (Invitrogen) baculovirus expression vector. The cloning of the hBLT1-WT plasmid and expression in *Sf9* insect cells for binding assays were carried out using the same methods as that of the hBLT1-CC. Expression for each plasmid was done in 250 mL volumes of biomass. Frozen biomass was thawed and resuspended in a cell lysis buffer (180 mM sucrose, 5 mM MgCl₂, 20 mM HEPES pH 7.3, and in-house protease inhibitor cocktail). Membrane fractions were isolated by repeated Dounce homogenization and low-speed centrifugation for 12 min at 4 °C and 300×g in cell lysis buffer (three times), keeping the supernatants in each round. The pooled supernatants underwent ultracentrifugation for 45 min at 4 °C and 210,000×g. The supernatant was discarded, and the pellet was resuspended and homogenized in membrane resuspension buffer (100 mM NaCl, 20 mM HEPES pH 7.3, and in-house protease inhibitor cocktail) to a final concentration of 1 mg mL⁻¹ cell membrane. Samples were aliquoted, flash-frozen in liquid nitrogen, and stored at −80 °C.

**Radioligand binding assays.** Membrane fractions isolated from *Sf9* insect cells expressing either hBLT1-WT or hBLT1 mutants were incubated at RT for 2 h with [³H]–LTB4 (ARC) in assay buffer (10 mM Tris-HCl pH 7.4, 10 mM CaCl₂, 10 mM MgCl₂) in a total assay volume of 100 μL. The unbound ligand was removed by rapid filtration through GF/C glass fiber filters and 3 × 3 mL washes with 40 mM HEPES pH 7.4 and 0.2 % (w/v) CHAPS. Bound radioactivity was measured through liquid scintillation using Eco-Lume Liquid Scintillation Cocktail (MP Biomedicals) and detected using a Tri-Carb 2910TR liquid scintillation counter (Perkin Elmer). Competition studies were carried out by incubating membranes (20 μg total protein per well) with a range of concentrations of LTB4 and MK-D-046 (30 μM–0.007 nM) and with [³H]–LTB4 at 4 nM.

**hBLT2 cAMP assay.** The inhibitory potency of MK-D-046 at hBLT2 was determined using a cAMP dynamic assay kit (Cisbio, 62AM4PEC) as previously described[24]. Briefly, HEK293 cells (ATCC, CRL-1573) overexpressing hBLT2 were cultured in DMEM (ThermoFisher, 11995-065), 10% dialyzed FBS (ThermoFisher, 26400-036), 1× NEAA (ThermoFisher, 11140-050), 200 μg mL⁻¹ hygromycin (ThermoFisher, 10687-010), and 10 μg mL⁻¹ blasticidin (ThermoFisher, A11139). HBSS (Hyclone, SH 30268.01) with 20 mM HEPES (Gibco, 15630-106), 800 μM IBMX (Sigma, 15879), and 0.1% DTPA-purified BSA (Perkin Elmer, CR84-100) was the assay medium. The concentration of MK-D-046 ranged from 0.5 nM to 20 μM, resulting in a maximal residual DMSO concentration of 0.25%. After an incubation for 20 min at 37 °C, cells were stimulated with 5 nM 12(S)-HHTrE (Sigma, H1640) and 1.5 μM forskolin (Sigma, F-6886) (final concentration in 10 μL reaction volume). The levels of cAMP were detected using the 2-step CisBio kit following the manufacturer's instructions. A total of 10 data points was collected for each curve, $n = 2$. There was no inhibition detected for MK-D-046 at hBLT2.

**Docking to hBLT1 and gpBLT1 structures.** Crystal structures of hBLT1 determined in this work and gpBLT1 (PDB ID 5X33) were used for docking, which was

performed in ICM-Pro v. 3.8.7b (Molsoft LLC) molecular modeling software. The structures of the receptors were protonated and optimized using an ICM docking pipeline. Molecular models of the compounds were generated from two-dimensional representations and their three-dimensional geometry was optimized using MMFF-94 force field[64]. Docking simulations used biased probability Monte Carlo (BPMC) optimization[65] of the compound's internal coordinates in the pre-calculated grid energy potentials of the receptor. The grid potentials, while freezing the conformational state of the receptor, implicitly take into account some receptor flexibility by using "soft" van Der Waals potentials. To ensure exhaustive sampling of the ligand-binding pose, the thoroughness parameter was set to 15, and at least 5 independent docking runs were performed for each compound starting from a random conformation. The results of individual docking runs for each molecule were considered consistent if at least three of the five docking runs produced similar ligand conformations. The unbiased docking procedure did not use distance restraints or any other a priori derived information for the ligand–receptor interactions.

**Reporting summary**. Further information on research design is available in the Nature Research Reporting Summary linked to this article.

## Data availability

Data supporting the findings of this manuscript are available from the corresponding author upon reasonable request. A reporting summary for this Article is available as a Supplementary Information file. Coordinates and structure factors for hBLT1 in complex with MK-D-046 have been deposited in the Protein Data Bank (PDB) with the accession code 7K15. Raw diffraction images have been uploaded to the Zenodo data repository with the DOI data identifier "10.5281/zenodo.4450301" [https://doi.org/10.5281/zenodo.4450301]. The amino acid sequences for BLT1 and BLT2 receptors used in this study are available from the UniProt database under the accession numbers: Q15722 (hBLT1), Q9WTK1 (gpBLT1), O88855 (mBLT1), Q9R0Q2 (rBLT1), Q9NPC1 (hBLT2). Source data are provided with this paper.

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

## Acknowledgements

We thank Jeffrey Velasquez, Kelly Villers, Chris Hanson, Monisha Gonzales, and Andrii Ishchenko for technical support; Angela Walker for administrative support; and Raymond C. Stevens for advice on the manuscript. This research was supported in parts by the National Institute of General Medical Sciences grant R35 GM127086, the GPCR Consortium, and the Canadian Institutes of Health Research (CIHR) grant FDN-148413. É.B.-O. is the recipient of research fellowships from the CIHR (MFE-164740) and the Fonds de recherche du Québec – Santé (255989) and a member of the California NanoSystem Institute. P.S. is the recipient of a Tier 1 Canada Research Chair in Neurophysiopharmacology of Chronic Pain and a member of the FRQ-S-funded Centre de recherche du CHUS. Parts of this research were completed at the GM/CA beamline 23ID-B at the Advanced Photon Source (APS) in the Argonne National Laboratory, IL, USA. GM/CA@APS has been funded in whole or in part with Federal funds from the National Cancer Institute (ACB-12002) and the National Institute of General Medical Sciences (AGM-12006). This research used resources of the Advanced Photon Source, a U.S. Department of Energy (DOE) Office of Science User Facility operated for the DOE Office of Science by Argonne National Laboratory under Contract No. DE-AC02-06CH11357. The Eiger-16M detector at GM/CA-XSD was funded by NIH grant S10 OD012289.

## Author contributions

N.M. managed the entire project; designed and cloned constructs for structural and functional studies; purified, characterized, and optimized all hBLT1 constructs for structural studies; developed the purification procedure for the hBLT1 crystallization construct; screened all ligands for structural studies and selected MK-D-046; crystallized the hBLT1 receptor; designed and prepared all crystallization plates; prepared all crystal samples for diffraction quality screening and for data collection; collected all X-ray diffraction data; analyzed and integrated all data for the project; wrote the manuscript; and prepared all figures and tables for the manuscript. N.M. and V.C. performed data processing and structure determination. N.M. and G.W.H. performed structure refinement and quality control. A.S. performed molecular docking studies. V.K. supervised molecular docking studies. É.B.-O. performed functional assays. P.S. supervised functional assays. B.A.Z. performed radioligand binding assays. H.K., X.F., P.S., J.P., K.B.S., and S.M.S. selected and synthesized ligands for structural studies. V.K. and P.P. provided construct design suggestions. V.C. supervised the overall project. All authors edited and approved the manuscript.

## Competing interests

H.K., B.A.Z., X.F., P.S., J.P., K.B.S and S.M.S. are employees of Merck & Co. The remaining authors declare no competing interests.
