## [Peer Review File · Nature Communications]

REVIEWER COMMENTS

Reviewer #1 (Remarks to the Author):

In this paper, the authors describe the crystal structure of human leukotriene B4 receptor 1 (hBLT1) solved at a 2.9 Å resolution. This structure provides the general organization of the receptor, gives the architecture of the site responsible for binding their antagonist and some hints on the way the natural ligand may access its binding site. In addition, the authors compare their structure to that of a guinea pig ortholog (PgBLT1) previously solved at a lower resolution and then carry out a series of docking experiments to demonstrate that their structure is adapted to predict the binding of ligands.

Although incremental compared to the previous structure of PgBLT1 published in 2018 in Nature Chem. Biol., this is an interesting report. As the authors claim, their better resolution as well as dealing with a human ortholog represents a step forward, in particular in a drug-design perspective. As such, this paper deserves publication.

In its present state, however, this paper leaves the reader with the impression that the authors present their structure and then provide an overview of the different aspects of receptor functioning - ligand binding process, selectivity between different receptor subtypes, sodium binding site...- that are supported by limited set of experimental evidences besides the structure itself. In my opinion, experimentally strengthening several of their conclusion would take the paper a step forward. For instance:

- All their conclusions on the role of specific residues are based on functional assays. However, such assays can in some cases be confusing, as they are the result of several processes (ligand binding, receptor activation, coupling to the signaling partner). As such, they can in some cases lead to misleading conclusions. For instance, one could conclude that the ICL3-flav does not bind the ligands anymore, although the authors state that they observed no IP1 production because the fusion partner prevents coupling to the G protein partner, which is certainly the case. In the same way, some mutations in the BLT1 receptor were shown to lead to loss of signaling without dramatically affecting binding (e.g. Babu et al. (2007)). The authors should confirm their conclusions through direct ligand binding experiments rather than functional assays, as this is important in particular when dealing with residues that are proposed to be directly responsible for ligand recognition.

- the authors conclude on the existence of a LTB4 entry channel based on the mutation of very few residues that may line up this possible channel. However, in my opinion, its difficult to assess this point firmly at the present stage. Indeed, it is difficult to make a difference between an effect due to the binding itself or to the entry process based on this sole evidence. In addition, in their structure the ligand binding pocket is widely exposed to the solvent and not occluded, as it is the case in some structures of lipid-binding GPCRs, including that of gpBLT1 where it is partly blocked. The authors should go further if they really want to address this point. Indeed, this a really important aspect of receptor functioning that may deserve further investigation.

- The authors make a thorough comparison of their structure and that of pgBLT1. This is important, in particular in a drug-design perspective where ligands are tested in animal models and this can lead to misleading conclusions when using them in human systems. The authors point to the differences in the architecture of the ligand binding pocket of hBLT1 and pgBLT1 and state that this may be the combined result of the h/pg and resolution differences. Could such differences also arise from the differences in the kind of ligand used ? The authors tested the impact of some "reverse" mutations on the binding of their MK-D-046 ligand, but did they try to analyze how the features they inferred from their structure impact on BIIL-260 binding, as this ligand binds the human receptor as well? In the

same way, amino-acid substitutions have very modest impact on the potency of MK-D-046 (2-fold maximum effect). The authors state that "when combined, all four mutations could produce much stronger effects...". It would be important to verify it to really assess their conclusion on a firm basis.

- the difference in BLT1 and BLT2 is an important aspect of the LTB4 system and could lead to selective ligands for therapeutic applications; the authors attribute the differences in specificity between BLT1 and BLT2 receptors to a very limited set of residues that when mutated to the equivalent amino-acids of BLT2 lead to a dramatic decrease in LTB4 potency. Again, as stated above, it would be interesting to assess whether such mutations affect LTB4 recognition rather than receptor activation, as BLT2 displays a significantly reduced affinity for LTB4. Besides, could the authors propose some experiments to really strengthen their conclusion, such as making the reverse mutations in BLT2 and testing ligands that are more specific of BLT2 ?

Minor points:

- The authors give a sentence on the H8 helix. This is nevertheless an interesting feature of their structure as this region is not visible in the previous crystal structure of gpBLT1 and that H8 is an important component of BLT1 functioning (see e.g. Aratake et al. (2012) FASEB J.). Is the positioning of H8 really an artifact of their construct, as they suggest, or could it have some implications for instance on receptor recycling ? please comment, if possible.

- The authors indicate the possible implications of their crystal packing for BLT1 dimerization. This is highly speculative, in my opinion. The authors should either strengthen their point or remove the sentence.

- More than half of the discussion deals with the interest of the BLT1 system for treating diabetes... This is a very interesting topic, but it nevertheless seems to me that it lies rather far from the implications of their work and would be more adapted to a general review. Maybe the authors could shorten this general part of the discussion to focus a little bit more on the implications of their work for the leukotriene signaling field.

Reviewer #2 (Remarks to the Author):

The manuscript submitted by Michaelian et al. reports the 2.9Å resolution crystal structure of the human Leukotriene B4 Receptor 1 (hBLT1) in complex with MK-D-046, a selective antagonist of medicinal relevance in the context of inflammatory diseases and type 2 diabetes (T2D). The structural work is complemented with site-directed mutagenesis, docking studies, as well as a comparison with the guinea pig BLT1 structure obtained at 3.7Å resolution in 2018. By identifying critical residues, this work contributes to a better understanding of BLT1 binding modes and provides insights into the developments of therapeutics against inflammatory diseases and T2D in particular. This manuscript is appropriate for publication in Nature Communications. Some comments are listed below:

P4 l.89 - Add "respectively".

P5 l.94 - Which residues are responsible for the dimerization? Are they the same as in Baneres et al.?

P9 l.194 / Fig 3f - Please indicate in Fig 3f where the Na site is located.

P9 I.198 and Sup Fig 3h - Sup Fig 3h shows a nice globular blob. Why is it not possible to differentiate Na from benzamidine? At 2.9Å resolution, it should be clear.

Sup table 1 - Please report statistics for the inner shell.

Sup table 1 - Please indicate which completeness is reported, spherical or ellipsoidal?

If the 84.9% reported value corresponds to the ellipsoidal completeness (as it should be), then it is likely that the crystallographic data collection suffered from having systematically measured the 32 crystals in roughly the same orientation. With this amount of crystals and in P21221 space group, one should be able to report close to 100% completeness at low resolution, even with the use of the program STARANISO. Otherwise, more details on the data collection protocol should be given.

P10 I.226 - a word is missing after "we"

P20 I.448 - remove "approximately" and state the range of frames collected.

P24 I.547 - Please make the raw crystallographic data available.

Reviewer #3 (Remarks to the Author):

Michaelian et al have determined the structure of the human leukotriene B4 receptor (hBLT1) in the inactive state bound to a potent antagonist. The structure is of good quality giving a high degree of confidence in the conclusions derived from its analysis. The structure was used for docking studies of related antagonists to determine a rationale for the observed SAR.

The structure presented is the second of BLT1, with the structure of guinea pig BLT1 (gpBLT1) being published in 2018. The current manuscript is important in a number of respects. The structure of hBLT1 is of significantly better resolution than that of gpBLT1 and contains additional ordered regions absent in the gpBLT1 structure. The ligands co-crystallised with the respective receptors are very different and although they both bind in the orthosteric binding pocket, the ligand bound to gpBLT1 extends into the Na⁺ ion binding pocket, unlike the ligand bound to hBLT1. Finally, there are differences in the amino acid sequence of gpBLT1 around the orthosteric binding pocket. The combination of the different sequence of gpBLT1 and hBLT1 and the different ligands, meant that docking of some antagonists to gpBLT1 that are known to bind effectively to hBLT1 was unsuccessful. Thus in terms of structure-based drug design to treat human diseases, the new structure has considerable advantages over the previous structure from guinea pig. The authors also performed docking studies with a native agonist LTB4 into the inactive state and identified two potential binding modes that will further research into BLT1 receptor activation.

The manuscript is extremely well written, with clear figures and well-presented data. There are only a few minor points that need addressing.

P3, para 1 lines 15-18. The sentence starting 'Due to differences...' is a discussion point arising from the structure determination of hBLT1. It is therefore inappropriate in the introduction and should be deleted. This point is discussed at length in the results section.

P4, para 1: In the results section, on the first mention of LTB4 (line 7), please describe it as 'the agonist LTB4' and similarly use the term 'the antagonist MK-D-046' (line 15) to help the reader unfamiliar with these ligands. In the same paragraph (line 8) I do not understand the phrase 'restores the conserved DRY motif'; reformulate.

P10, para 3, line 8; change 'we' to 'were'

P12 para 2, line 10 onwards. The way I understood the sentence on the mutation H181W was that the Trp blocked the entrance channel and therefore decreases ligand affinity. However, if the Trp only blocks the channel and does not interact with the ligand, then surely the on-rate and off-rate of the ligand will be affected equally and therefore the affinity will remain unchanged. Is there a potential interaction between H181 and the docked LTB4 and Trp blocks it from assuming a preferred orientation? A little bit of clarification would be helpful.

P16 para 3 lines 1-2. Please define the method for optimization of expression in insect cells
P18, para 1 line 3 and para 2 line 7; please define centrifugation speed, time and temperature.

P19 line 2, please define PSP.

Table 1. the n values for the Emax and Imax data are not given.

Supplementary Fig 3, panel g, please define red and green densities.

We would like to thank Reviewers for their helpful comments and constructive critique, which we fully addressed below. The manuscript was revised accordingly, and all changes are highlighted in yellow. Our point-by-point responses to the Reviewers' comments are shown below in bold:

REVIEWER COMMENTS

Reviewer #1 (Remarks to the Author):

In this paper, the authors describe the crystal structure of human leukotriene B4 receptor 1 (hBLT1) solved at a 2.9 Å resolution. This structure provides the general organization of the receptor, gives the architecture of the site responsible for binding their antagonist and some hints on the way the natural ligand may access its binding site. In addition, the authors compare their structure to that of a guinea pig ortholog (PgBLT1) previously solved at a lower resolution and then carry out a series of docking experiments to demonstrate that their structure is adapted to predict the binding of ligands.

Although incremental compared to the previous structure of PgBLT1 published in 2018 in Nature Chem. Biol., this is an interesting report. As the authors claim, their better resolution as well as dealing with a human ortholog represents a step forward, in particular in a drug-design perspective. As such, this paper deserves publication.

In its present state, however, this paper leaves the reader with the impression that the authors present their structure and then provide an overview of the different aspects of receptor functioning - ligand binding process, selectivity between different receptor subtypes, sodium binding site...- that are supported by limited set of experimental evidences besides the structure itself. In my opinion, experimentally strengthening several of their conclusion would take the paper a step forward. For instance:

- All their conclusions on the role of specific residues are based on functional assays. However, such assays can in some cases be confusing, as they are the result of several processes (ligand binding, receptor activation, coupling to the signaling partner). As such, they can in some cases lead to misleading conclusions. For instance, one could conclude that the ICL3-flav does not bind the ligands anymore, although the authors state that they observed no IP1 production because the fusion partner prevents coupling to the G protein partner, which is certainly the case. In the same way, some mutations in the BLT1 receptor were shown to lead to loss of signaling without dramatically affecting binding (e.g. Babu et al. (2007)). The authors should confirm their conclusions through direct ligand binding experiments rather than functional assays, as this is important in particular when dealing with residues that are proposed to be directly responsible for ligand recognition.

We performed radioligand binding assays to evaluate the affinity of LTB4 and MK-D-046 at hBLT1-WT and hBLT1-CC constructs, as well as H94^{3.29}Y, R156^{4.64}K, I271^{7.39}T, and H181^{5.38}W mutations. Binding assays confirm that H94^{3.29}, R156^{4.64}, and I271^{7.39} are critical

residues for ligand recognition. These new data are presented in Supplementary Table 1 and Supplementary Fig. 3. The text has also been updated to include the new binding data.

- the authors conclude on the existence of a LTB4 entry channel based on the mutation of very few residues that may line up this possible channel. However, in my opinion, its difficult to assess this point firmly at the present stage. Indeed, it is difficult to make a difference between an effect due to the binding itself or to the entry process based on this sole evidence. In addition, in their structure the ligand binding pocket is widely exposed to the solvent and not occluded, as it is the case in some structures of lipid-binding GPCRs, including that of gpBLT1 where it is partly blocked. The authors should go further if they really want to address this point. Indeed, this a really important aspect of receptor functioning that may deserve further investigation.

We agree with the Reviewer that this question is important and that it is very difficult to establish the entrance route for LTB4 into the binding pocket experimentally. Kinetic radioligand disassociation assays similar to those that we did for melatonin receptors (Johansson et al 2019, Nature 569, 289) would be a possibility, however, they are complicated in case of BLT1 by the lipid radioligand that can partition in the lipid membrane. We have been very careful with not overstating the membrane channel and emphasizing that while the proposed membrane entrance is consistent with the type of the endogenous ligand, the shape of the binding pocket, and the effect of several mutations, it would require a whole dedicated study to fully test this hypothesis.

For this revision, we evaluated the putative channel-blocking mutation H181^{5.38}W in radioligand binding assays and found that it has no effect on LTB4 affinity. This result is expected, as blocking the channel entrance should only affect the ligand on and off rates but not the equilibrium binding constant.

- The authors make a thorough comparison of their structure and that of pgBLT1. This is important, in particular in a drug-design perspective where ligands are tested in animal models and this can lead to misleading conclusions when using them in human systems. The authors point to the differences in the architecture of the ligand binding pocket of hBLT1 and pgBLT1 and state that this may be the combined result of the h/pg and resolution differences. Could such differences also arise from the differences in the kind of ligand used ? The authors tested the impact of some “reverse” mutations on the binding of their MK-D-046 ligand, but did they try to analyze how the features they inferred from their structure impact on BIIL-260 binding, as this ligand binds the human receptor as well? In the same way, amino-acid substitutions have very modest impact on the potency of MK-D-046 (2-fold maximum effect). The authors state that “when combined, all four mutations could produce much stronger effects...”. It would be important to verify it to really assess their conclusion on a firm basis.

We completed a new set of functional assays where we evaluated hBLT1-4 mut construct containing all 4 reverse gpBLT1 mutations (F169^{ECL2}L, P170^{ECL2}A, S264^{7.32}R, and N268^{7.36}K) as well as gpBLT1-WT. These constructs were evaluated for their impact on potency and efficacy in the presence of MK-D-046 and BIIL-260 (hBLT1-WT was also evaluated with BIIL-260).

In case of both MK-D-046 and BIIL-260, 4 reverse mutations do not fully explain the differences in potency of these ligands at hBLT1 and gpBLT1. We propose that variations in residues outside the ligand-binding pocket are mainly responsible for the observed effects. These new data are presented in Table 1, Supplementary Table 4, and Supplementary Figs. 1e and 2e,f. The text has also been updated to reflect the new data.

- the difference in BLT1 and BLT2 is an important aspect of the LTB4 system and could lead to selective ligands for therapeutic applications; the authors attribute the differences in specificity between BLT1 and BLT2 receptors to a very limited set of residues that when mutated to the equivalent amino-acids of BLT2 lead to a dramatic decrease in LTB4 potency. Again, as stated above, it would be interesting to assess whether such mutations affect LTB4 recognition rather than receptor activation, as BLT2 displays a significantly reduced affinity for LTB4. Besides, could the authors propose some experiments to really strengthen their conclusion, such as making the reverse mutations in BLT2 and testing ligands that are more specific of BLT2?

In the new radioligand binding assays, hBLT1 mutants H94^{3.29}Y and I271^{7.39}T were evaluated in the presence of LTB4 and MK-D-046. Both mutations are reverse hBLT2 mutations. H94^{3.29}Y caused ~870-fold decrease in MK-D-046 affinity, and I271^{7.39}T caused >20-fold decrease in LTB4 affinity, indicating that both residues are not only important for ligand recognition but may also impact differences in subtype selectivity. The new binding data are presented in Supplementary Table 1 and Supplementary Fig. 3c,e.

We have also completed a new set functional assays, that test hBLT2-WT and reverse hBLT2 mutants, Y129^{3.29}H and T305^{7.39}I, with a BLT2-selective agonist 12S-HHTrE and BLT2-selective antagonist LY-255283. The first mutation, Y129^{3.29}H, had little effect on both agonist and antagonist, suggesting that this residue does not interact with BLT2-selective ligands. The second mutation decreased antagonist potency by ~5-fold, indicating that T305^{7.39} potentially contributes to BLT2 selectivity of antagonists. The new hBLT2 functional data are presented in Supplementary Table 7 and Supplementary Fig. 9.

Minor points:

- The authors give a sentence on the H8 helix. This is nevertheless an interesting feature of their structure as this region is not visible in the previous crystal structure of gpBLT1 and that H8 is an important component of BLT1 functioning (see e.g. Aratake et al. (2012) FASEB J.). Is the positioning of H8 really an artifact of their construct, as they suggest, or could it have some implications for instance on receptor recycling? please comment, if possible.

We believe that the secondary structure of H8 is correctly captured, however, the orientation of H8 in the hBLT1 crystal structure is potentially affected by crystal contacts and/or by the artificial sequence that succeeds H8 and consists of the EcoRI + PreScission Protease site (EFLEVLQ). The artificial sequence forms a small α -helix that folds on the

intracellular side of the receptor making interactions with several TMs and intracellular loops.

- The authors indicate the possible implications of their crystal packing for BLT1 dimerization. This is highly speculative, in my opinion. The authors should either strengthen their point or remove the sentence.

The sentence has been removed, and we have clarified that it is a crystallographic parallel dimer interface.

- More than half of the discussion deals with the interest of the BLT1 system for treating diabetes... This is a very interesting topic, but it nevertheless seems to me that it lies rather far from the implications of their work and would be more adapted to a general review. Maybe the authors could shorten this general part of the discussion to focus a little bit more on the implications of their work for the leukotriene signaling field.

The discussion was modified to include a brief and general review on targeting BLT1 for treating diabetes. A more thorough discussion on hBLT1/hBLT2 selectivity and signaling as well as hBLT1/gpBLT1 selectivity was added as these were significant points in the manuscript and required more elaboration, especially due to results from the newly added functional and binding assays.

Reviewer #2 (Remarks to the Author):

The manuscript submitted by Michaelian et al. reports the 2.9Å resolution crystal structure of the human Leukotriene B4 Receptor 1 (hBLT1) in complex with MK-D-046, a selective antagonist of medicinal relevance in the context of inflammatory diseases and type 2 diabetes (T2D). The structural work is complemented with site-directed mutagenesis, docking studies, as well as a comparison with the guinea pig BLT1 structure obtained at 3.7Å resolution in 2018. By identifying critical residues, this work contributes to a better understanding of BLT1 binding modes and provides insights into the developments of therapeutics against inflammatory diseases and T2D in particular.

This manuscript is appropriate for publication in Nature Communications. Some comments are listed below:

P4 1.89 - Add "respectively".

The requested change was made in the revised manuscript.

P5 1.94 - Which residues are responsible for the dimerization? Are they the same as in Baneres et al.?

This section has been modified. In our hBLT1 structure, TM1 and H8 are responsible for creating parallel contacts in the crystal packing. In Baneres et al. (Structure-based analysis of GPCR function: evidence for a novel pentameric assembly between the dimeric leukotriene B4 receptor BLT1 and the G-protein, J. Mol. Biol., 2003), TM6 of BLT1 is responsible for dimerization. The Baneres et al. study was done with detergent-solubilized receptor, which can yield different results from receptors in LCP.

However, because we cannot be certain about the functional implications of H8 due to the succeeding artificial sequence, we have adjusted this section to make it clear that it is a crystallographic parallel dimer.

P9 1.194 / Fig 3f - Please indicate in Fig 3f where the Na site is located.

Fig. 3f has been modified to indicate the location of the sodium (Na⁺) site. The figure legend has also been modified.

P9 1.198 and Sup Fig 3h - Sup Fig 3h shows a nice globular blob. Why is it not possible to differentiate Na from benzamidine? At 2.9Å resolution, it should be clear.

After a closer evaluation of the structure and additional rounds of refinement, we have modeled a sodium ion (Na⁺) in the sodium site (now Sup Fig 4h). Benzamidine was ruled out as it would clash with MK-D-046 if it would bind to the sodium site in the same position as seen with BIIL-260 in gpBLT1. Other figures have also been modified to indicate the new version of the hBLT1 structure that shows the modeled Na⁺.

Sup table 1 - Please report statistics for the inner shell.

The statistics for the inner (lowest resolution) shell are typically not reported in Table 1. However, we have provided them here:

Resolution: 40.00-6.24

R_{merge}: 0.127

I/σI: 12.4

Completeness (%): 98.5

Redundancy: 9.9

Sup table 1 - Please indicate which completeness is reported, spherical or ellipsoidal?

If the 84.9% reported value corresponds to the ellipsoidal completeness (as it should be), then it is likely that the crystallographic data collection suffered from having systematically measured the 32 crystals in roughly the same orientation. With this amount of crystals and in P21221 space group, one should be able to report close to 100% completeness at low resolution, even with the

use of the program STARANISO. Otherwise, more details on the data collection protocol should be given.

The reported completeness is spherical, since it was calculated before applying ellipsoidal truncation. The overall ellipsoidal completeness is 95.3% (97.4% in the lowest resolution shell) as reported by STARANISO. There are no significant systematically missed wedges of data.

P10 1.226 - a word is missing after "we"

We thank Reviewer for noticing this typo. We changed “we” to “were” in the revised version.

P20 1.448 - remove "approximately" and state the range of frames collected.

We replaced “approximately” with the range of frames collected per each crystal.

P24 1.547 - Please make the raw crystallographic data available.

Raw diffraction images have been uploaded to Zenodo data repository under the DOI number 10.5281/zenodo.4450301 (<https://doi.org/10.5281/zenodo.4450301>), with the title ‘Raw Diffraction Data of the Human Leukotriene B4 Receptor 1 in Complex with Antagonist MK-D-046’ .

Reviewer #3 (Remarks to the Author):

Michaelian et al have determined the structure of the human leukotriene B4 receptor (hBLT1) in the inactive state bound to a potent antagonist. The structure is of good quality giving a high degree of confidence in the conclusions derived from its analysis. The structure was used for docking studies of related antagonists to determine a rationale for the observed SAR.

The structure presented is the second of BLT1, with the structure of guinea pig BLT1 (gpBLT1) being published in 2018. The current manuscript is important in a number of respects. The structure of hBLT1 is of significantly better resolution than that of gpBLT1 and contains additional ordered regions absent in the gpBLT1 structure. The ligands co-crystallised with the respective receptors are very different and although they both bind in the orthosteric binding pocket, the ligand bound to gpBLT1 extends into the Na⁺ ion binding pocket, unlike the ligand bound to hBLT1. Finally, there are differences in the amino acid sequence of gpBLT1 around the orthosteric binding pocket. The combination of the different sequence of gpBLT1 and hBLT1 and the different ligands, meant that docking of some antagonists to gpBLT1 that are known to bind effectively to hBLT1 was unsuccessful. Thus in terms of structure-based drug design to

treat human diseases, the new structure has considerable advantages over the previous structure from guinea pig. The authors also performed docking studies with a native agonist LTB4 into the inactive state and identified two potential binding modes that will further research into BLT1 receptor activation.

The manuscript is extremely well written, with clear figures and well-presented data. There are only a few minor points that need addressing.

P3, para 1 lines 15-18. The sentence starting ‘Due to differences...’ is a discussion point arising from the structure determination of hBLT1. It is therefore inappropriate in the introduction and should be deleted. This point is discussed at length in the results section.

This sentence was deleted as requested.

P4, para 1: In the results section, on the first mention of LTB4 (line 7), please describe it as ‘the agonist LTB4’ and similarly use the term ‘the antagonist MK-D-046’ (line 15) to help the reader unfamiliar with these ligands. In the same paragraph (line 8) I do not understand the phrase ‘restores the conserved DRY motif’; reformulate.

The terms ‘the agonist LTB4’ and ‘the antagonist MK-D-046’ are now included in the text.

The DRY motif is a motif commonly found in transmembrane helix 3 (TM3) in class A GPCRs. We clarified this in the text and also included citations for support. Since in hBLT1-WT this motif is represented as DRS, ‘restore’ is used to refer to the fact the mutation S116^{3.51}Y returns the motif back to its most conserved form.

P10, para 3, line 8; change ‘we’ to ‘were’

The change was made in the revised manuscript.

P12 para 2, line 10 onwards. The way I understood the sentence on the H181W mutation was that the Trp blocked the entrance channel and therefore decreases ligand affinity. However, if the Trp only blocks the channel and does not interact with the ligand, then surely the on-rate and off-rate of the ligand will be affected equally and therefore the affinity will remain unchanged. Is there a potential interaction between H181 and the docked LTB4 and Trp blocks it from assuming a preferred orientation? A little bit of clarification would be helpful.

Our initial hypothesis was that this mutation would either slow down LTB4 binding kinetics substantially so that it would manifest as an apparent drop in potency in the IP₁ production assay, or it would directly interfere with the bound ligand. In this revision, we measured the affinity of LTB4 at the H181^{5.38}W mutant and found that it remains the same as at the WT receptor, confirming that W181^{5.38} has no direct interactions with LTB4. Changes were made to better clarify this in the text.

P16 para 3 lines 1-2. Please define the method for optimization of expression in insect cells

The optimization refers to the codon optimization of the hBLT1-WT sequence for insect cell expression. This was clarified in the text by adding the word ‘codon’.

P18, para 1 line 3 and para 2 line 7; please define centrifugation speed, time and temperature.

Details on centrifugation were added to the text.

P19 line 2, please define PSP.

PSP is PreScission Protease, and the change was made in the text.

Table 1. the n values for the Emax and Imax data are not given.

The n values for Emax and Imax are the same as the corresponding n values for EC₅₀ and IC₅₀. They have been now explicitly listed in Table 1.

Supplementary Fig 3, panel g, please define red and green densities.

Red and green densities have been defined. Supplementary Fig 3g is now Supplementary Fig. 4g.

REVIEWERS' COMMENTS

Reviewer #1 (Remarks to the Author):

The authors appropriately addressed all my concerns, and I have no additional comment on this interesting work.

Reviewer #2 (Remarks to the Author):

The authors have properly addressed the Reviewers' comments in their revised manuscript, which can now be accepted for publication.

Reviewer #3 (Remarks to the Author):

The authors have included extra information on ligand binding assays and mutants to substantiate the claims made in their previous version. The data are robust and support the conclusions of the authors. It was not easily possible to address the on/off rates of the ligand due to the hydrophobicity of the ligand, so these experiments were not performed, but this does not detract from the overall message of the manuscript.